# Learning Deliberately, Acting Intuitively: Unlocking Test-Time Reasoning in Multimodal LLMs

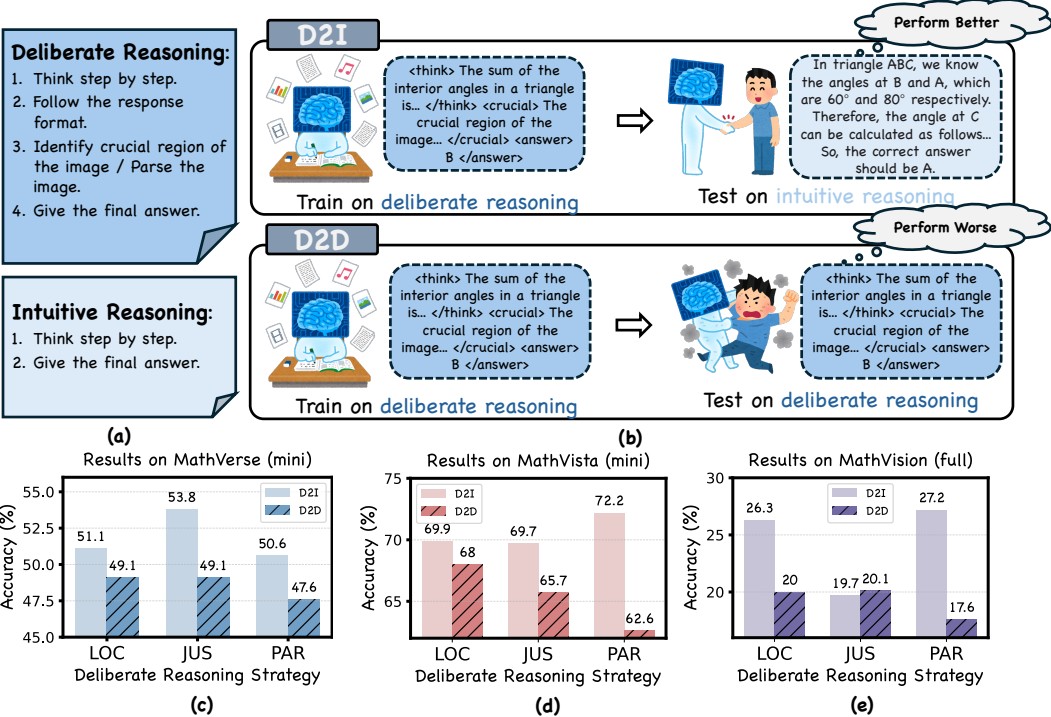

Figure 1: **(a)** The concept of our designed deliberate reasoning and intuitive reasoning. **(b)** The training and testing paradigm of our proposed deliberate-to-intuitive (D2I) framework. We refer to the commonly used training-testing framework in other works as D2D. **(c)(d)(e)** The performance on MathVerse, Mathvista and MATH-Vision. Our D2I with different deliberate reasoning strategies consistently outperforms D2D, illustrating that the reasoning ability is unlocked during test time.

## Abstract

Reasoning is a key capability for large language models (LLMs), particularly when applied to complex tasks such as mathematical problem solving. However, multimodal reasoning research still requires further exploration of modality alignment and training costs. Many of these approaches rely on additional data annotation and relevant rule-based rewards to enhance the understanding and reasoning ability, which significantly increases training costs and limits scalability. To address these challenges, we propose the **D**eliberate-to-**I**ntuitive reasoning framework (D2I) that improves the understanding and reasoning ability of multimodal LLMs (MLLMs) without extra annotations and complex rewards. Specifically, our method sets deliberate reasoning strategies to enhance modality alignment only through the rule-based format reward during training. While evaluating, the reasoning style shifts to intuitive, which removes deliberate reasoning strategies during training and implicitly reflects the model's acquired abilities in the response. D2I outperforms

baselines across both in-domain and out-of-domain benchmarks. Our findings highlight the role of format reward in fostering transferable reasoning skills in MLLMs, and inspire directions for decoupling training-time reasoning depth from test-time response flexibility.

# 1 INTRODUCTION

Recently, a growing body of research has distinguished between fast-thinking models (Cheng et al., 2025b; Li et al., 2025), which rely on immediate pattern recognition, and slow-thinking models (Dou et al., 2025), which explicitly simulate step-by-step reasoning. As the requirements of large language models (LLMs) (Zhang et al., 2024a) increase for complex tasks, such as mathematical problem-solving, their ability to perform long reasoning processes and slow thinking becomes a key factor in achieving reliable performance. Recent works, such as OpenAI's o1 series (Jaech et al., 2024), adopt the slow-thinking paradigm by scaling up inference-time Chain-of-Thought (CoT) (Xia et al., 2024) reasoning, yielding notable improvements in tasks such as mathematics and code generation. After that, DeepSeek-R1 (Guo et al., 2025) demonstrates that reinforcement learning (RL) (Chen et al., 2025b) can further enhance slow-thinking behavior by optimizing reasoning-specific objectives beyond standard supervised learning.

However, despite their significant advancements, these models still suffer from limited multimodal reasoning performance. Two major challenges underlie this limitation. The first is **multimodal alignment**: Accurate alignment serves as a fundamental prerequisite for reasoning in multimodal LLMs (MLLMs) (Zheng et al., 2024; Dai et al., 2023). In complex tasks such as multimodal mathematics (Wang et al., 2024), where fine-grained visual clues must be correctly identified, any misalignment between modalities undermines the model's ability to reason meaningfully. The second challenge concerns the **training efficiency and scalability of reasoning supervision**. Many recent approaches (Peng et al., 2024; Yang et al., 2025) rely on additional data annotation or complex rule-based rewards to guide models toward better reasoning behavior. While effective, these strategies significantly increase training costs and limit scalability, particularly in multimodal settings where annotation is both expensive and ambiguous. As a result, the development of lightweight and generalizable training strategies for multimodal reasoning remains an open and important direction.

To address these challenges, our goal is to develop an R1-style visual reasoning model that excels in complex, multimodal reasoning tasks. Specifically, we aim to equip MLLMs with stronger multimodal understanding and reasoning abilities through lightweight training strategies. Inspired by the distinction between training and inference phases, we observe that while the training process should encourage the model to acquire new capabilities (*e.g.*, understanding and analyzing images) through aggressive supervision and relatively smaller response search space, the inference process should focus on allowing a larger search space to explore more effective solutions by imposing minimal constraints (*e.g.*, achieving the training objectives implicitly), thereby producing reliable and high-confidence responses with the acquired capabilities.

To this end, we propose the **D**eliberate-**to**-**I**ntuitive (**D2I**) reasoning framework as shown in Figure 1, which enforces deep, deliberate reasoning behaviors during training and allows more flexible, intuitive responses at test time. Additionally, we refer to the setting where both the training and inference stages are characterized by deliberate reasoning behaviors as D2D, as is commonly adopted in most existing works (Zhao et al., 2025b; Zhang et al., 2023). D2I adopts a similar strategy to Group Relative Policy Optimization (GRPO) (Guo et al., 2025), employing simple rule-based rewards to guide model optimization. However, unlike GRPO, our method introduces additional deliberate reasoning strategies (*e.g.*, identifying the coordinates of a crucial region in the image or parsing visual structure) using corresponding response format constraints, which aims to allow the model to more thoroughly capture the semantic content of the image. These objectives promote deliberate and deep reasoning without requiring extra human annotations. At inference time, the model is evaluated under intuitive reasoning setting, where we remove deliberate reasoning strategies in the training and model's acquired abilities are implicitly reflected in the response. *Our findings suggest that only applying rule-based format supervision for deliberate reasoning strategies during training, without explicit content-level supervision, is sufficient to enhance the model's multimodal reasoning and understanding across both in-domain and out-of-domain benchmarks.*

Our contributions can be summarized as follows:

- We propose D2I, a reinforcement learning framework that enhances the reasoning ability of MLLMs by employing deliberate reasoning during training and intuitive reasoning during inference.
- To enhance the multimodal understanding ability for better reasoning, corresponding deliberate reasoning strategies are designed during training. For efficient reasoning, we didn't annotate the dataset and only supervised the response format.
- Experiments on Qwen2.5-vl-7B demonstrate the effectiveness of our D2I on both in-domain and out-of-domain multimodal benchmarks.

## 2 RELATED WORK

**MLLMs.** Multimodal large language models (MLLMs) are an extension of large language models (LLMs) (Chung et al., 2022; Bai et al., 2023; Touvron et al., 2023; Li et al., 2025), enhancing their ability to process both textual and visual inputs. By combining the strong reasoning capabilities of LLMs with the rich visual features extracted by vision backbones, these models demonstrate advanced multimodal reasoning and deeper content understanding(Zhang et al., 2024a; Zhao et al., 2025a). Some approaches, such as LLaVA (Liu et al., 2023), NExT-GPT (Wu et al., 2023), and MiniGPT-v2 (Chen et al., 2023), use a linear projection layer to connect a frozen LLM with a visual encoder, enabling multimodal alignment. In contrast, other methods like InstructBLIP (Dai et al., 2023) and BLIP-2 (Li et al., 2023a) train a dedicated Q-Former projection module to bridge the gap between different modalities. Together, these developments showcase the rapid progress of MLLMs in tackling cross-modal reasoning and comprehension tasks.

**LLM Reasoning.** Reasoning in LLMs (Yang et al., 2025) refers to the ability to perform multi-step inference, often by simulating a process of deliberate thinking. This capability is particularly important for solving complex tasks such as mathematical problem solving (Tong et al., 2024), where simple pattern matching or shallow associations are insufficient. In this context, researchers have increasingly drawn inspiration from the psychological framework of fast thinking versus slow thinking (System 1 vs. System 2), where slow thinking represents a more reflective problem-solving process (Zavolokina et al., 2024). OpenAI's o1 models (Jaech et al., 2024) marked a significant milestone in this direction by introducing test-time scaling, where the reasoning process is extended via longer CoT prompting. DeepSeek-R1 (Guo et al., 2025) adopts a reinforcement learning framework that improves the model's reasoning ability. By optimizing rule-based rewards (Shao et al., 2024), R1 guides the model toward producing well-structured reasoning traces. Our work is inspired by these remarkable reasoning works and aims to further investigate a more appropriate and flexible training framework for MLLMs.

## 3 PRELIMINARY

### 3.1 PROBLEM DEFINITION

We focus on challenging multimodal reasoning tasks (Zhang et al., 2024b; Wang et al., 2024; Lu et al., 2023), particularly those involving mathematical problem solving that requires integrating both visual and textual information. In this setting, each input instance consists of a natural language question paired with a relevant image, such as a diagram, chart, or visualized equation. The goal is to produce an output that not only includes the final answer, but also provides a text-based reasoning process that reflects the intermediate steps leading to the solution.

Formally, given an input pair $x = (x^{\text{text}}, x^{\text{img}})$, where $x^{\text{text}}$ is the textual question and $x^{\text{img}}$ is the associated image, the model is expected to generate a response $y = (y^{\text{sol}}, y^{\text{ans}})$. Here, $y^{\text{sol}}$ denotes a coherent reasoning trace in natural language, and $y^{\text{ans}}$ is the final answer.

### 3.2 GRPO

GRPO (Guo et al., 2025) is an efficient variant of Proximal Policy Optimization (PPO) (Schulman et al., 2017), designed to reduce computational overhead while maintaining competitive learning effectiveness. The core idea of GRPO is to sample multiple responses for the same input and evaluate their relative quality. Instead of computing absolute rewards, the method calculates the groupwise

advantage by comparing each response to the average performance within the group. The objective function is:

$$\mathcal{J}_{GRPO} = \mathbb{E}[\frac{1}{N}\sum_{i}^{N} min(d_i A_i, clip(d_i, 1 - \epsilon, 1 + \epsilon)A_i - \beta \cdot \text{KL})] \tag{1}$$

Note that GRPO incorporates two types of rule-based rewards. The first is the format reward, which encourages the LLM to structure its response by enclosing the reasoning process within `<think></think>` and the final answer within `<answer></answer>`. The second is the answer reward, which rewards responses where the content enclosed within `<answer></answer>` matches the correct final answer. The final reward $r_i$ is computed as the average of these two components. In our settings, the reasoning process may also involve identifying or referencing specific visual elements (*e.g.*, bounding box coordinates or regions), requiring the model to ground its reasoning in the visual modality.

## 4 APPROACH

### 4.1 OVERVIEW

We propose the D2I reasoning framework to address the limitations of existing multimodal reasoning approaches. D2I is motivated by the observation that training-time reasoning can be more deliberate and structured for aggressive updating to enhance the understanding of multimodal input, while inference-time behavior should remain flexible and intuitive for stable performance. The key idea is to supervise deep reasoning behaviors during training, and allow unconstrained generation during inference. For the training-inference framework commonly adopted in most existing works, we refer to it as D2D, where both the training and inference stages are characterized by deliberate reasoning behaviors.

Within our framework, we design three types of deliberate reasoning strategies, each encouraging the model to engage in structured reasoning grounded in visual semantics: ***Region Localization (LOC) Strategy, Region Justification (JUS) Strategy and Parsing Consistency (PAR) Strategy***. These strategies target different aspects to enforce through rule-based format reward, without requiring additional human annotations. Details are shown as follows. While implementation, we keep the accuracy reward in GRPO and only adjust the corresponding format reward.

### 4.2 DELIBERATE REASONING STRATEGIES

**Region Localization (LOC) Strategy.** The LOC strategy encourages the model to identify which part of the image is most relevant to the reasoning process by explicitly outputting the coordinates (*e.g.*, bounding box) of that region. The corresponding format reward is triggered when a coordinate appears within the designated tag (*i.e.*, `<box>` $(5, 79), (110, 290)$ `</box>`) inside the reasoning trace. This objective guides the model to first locate before reasoning. As shown in Figure 2, we expect the model to generate responses in the following format:

```
<think> reasoning here <box> coordinate here </box> reasoning here </think>
<answer> final answer here </answer>
```

**Region Justification (JUS) Strategy.** The JUS strategy targets the explanation of crucial visual clues. Specifically, the model is encouraged to describe the image regions crucial for solving the question in natural language as part of its reasoning process. The format reward is assigned when the designated tag (*i.e.*, `<crucial>...</crucial>`) contains a coherent textual description that refers to visual elements. This supports interpretability and encourages visual-semantic alignment. As shown in Figure 2, we expect the model to generate responses in the following format:

```
<think> reasoning here </think> <crucial> textual explanation here </crucial>
<answer> final answer here </answer>
```

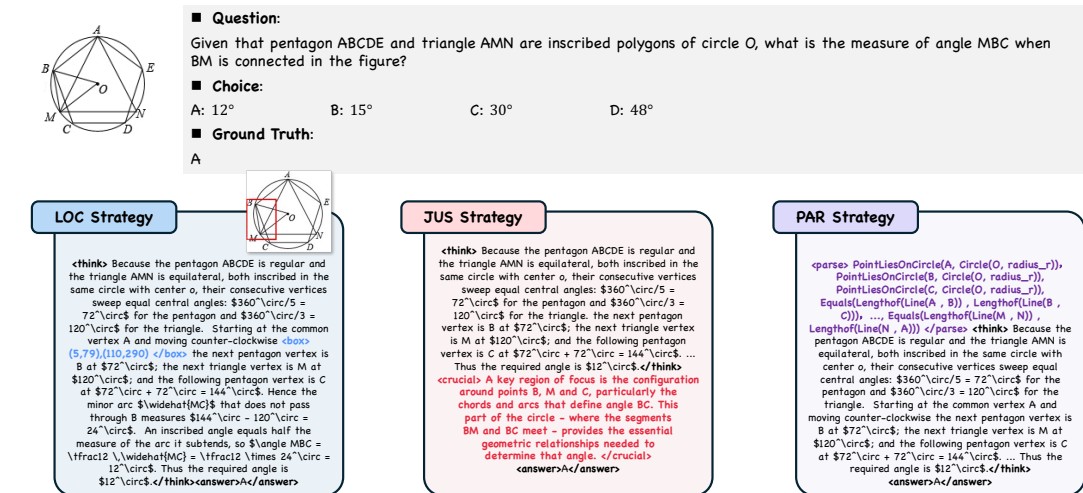

Figure 2: The response examples for our proposed three deliberate reasoning strategies during training.

**Parsing Consistency (PAR) Strategy.** The PAR strategy supervises the model to output a structured parsing of the input image before generating the reasoning trace. Parsing results, which convert the information in the image into the corresponding structured language or symbolic expression, are expected to appear in a predefined format (*i.e.*, a `<parse>...</parse>` block) at the beginning of the response. The format reward is given when the parsing result follows the correct format. This strategy promotes global visual understanding as a foundation for downstream reasoning. As shown in Figure 2, we expect the model to generate responses in the following format:

> `<parse>` *parsing result here* `</parse>` `<think>` *reasoning here* `</think>`
> `<answer>` *final answer here* `</answer>`

## 5 EXPERIMENT SETTINGS

### 5.1 DATASETS

**Training.** We follow the setup of R1V (Chen et al., 2025a) and use GEOQA-8K (Chen et al., 2025a) as our training dataset, which is a geometry-focused multimodal reasoning dataset consisting of text-image pairs that require mathematical and spatial understanding. The training split contains 8,030 examples.

**Evaluation.** We report the performance on the GEOQA-8K (Chen et al., 2025a) test set as in-domain performance, which includes 754 examples. To assess the generalization of our model, we also evaluate on several widely used out-of-domain benchmarks. Three of them are multimodal math benchmarks: MathVerse (Zhang et al., 2024b), MathVista (Lu et al., 2023), and MATH-Vision (Wang et al., 2024). These benchmarks encompass a diverse range of mathematical question types, enabling a comprehensive evaluation of the model's reasoning capabilities. The rest are multimodal benchmarks, including MME (Fu et al., 2024), MMVet (Yu et al., 2024), MMMU (Yue et al., 2024), SEED-Bench (Li et al., 2024a) and POPE (Li et al., 2023b), for general capability evaluation.

### 5.2 COMPARED METHODS

The methods include four aspects: (1) closed-source general models (*i.e.*, GPT-4V (OpenAI, 2023) and GPT-4o (Hurst et al., 2024)), (3) open-source general models (*i.e.*, Qwen2-VL-7B (Bai et al., 2023), InternVL2-8B (Chen et al., 2024b) and InternVL2.5-8B (Chen et al., 2024a)), (3) recent reasoning models (*i.e.*, LLaVA-CoT-11B (Xu et al., 2024), R1-Onevision-7B (Yang et al., 2025)

Table 1: Main results of our proposed GEOQA-8K-trained D2I method on both in-domain and out-of-domain test sets. The **bold** denotes the best performance, while the underline denotes the second best one. In the Qwen w/ GRPO, we evaluate the model with the deliberate reasoning, while in w/ GRPO$^\dagger$ we evaluate with the intuitive reasoning. $\Delta_{base}$ represents the improvement of D2I over Qwen2.5-VL-7B, while $\Delta_{grpo\dagger}$ represents the improvement over Qwen2.5-VL-7B w/ GRPO$^\dagger$. $^\star$ denotes our re-implementation.

| Method | In-domain | Out-of-domain (Math) | | | Out-of-domain (General) | | | | |
|---|---|---|---|---|---|---|---|---|---|
| | GEOQA-8K | MathVerse (mini) | MathVista (mini) | MATH-Vision (full) | MME (sum) | MMVet (turbo) | MMMU (val) | SEED | POPE |
| *Closed-Source General Models* | | | | | | | | | |
| GPT-4V (OpenAI, 2023) | – | 39.4 | 58.1 | 22.7 | 1926.6 | 67.5 | 63.1 | 53.8 | – |
| GPT-4o (Hurst et al., 2024) | – | 50.2 | 63.8 | **30.3** | – | 69.1 | **69.1** | 72.0 | 86.9 |
| *Open-Source General Models* | | | | | | | | | |
| Qwen2-VL-7B (Bai et al., 2023) | – | 31.9 | 58.2 | 16.3 | 2326.8 | 62.0 | 54.1 | 75.1 | 88.1 |
| InternVL2-8B (Chen et al., 2024b) | – | 37.0 | 58.3 | 18.4 | 2210.3 | 54.2 | 52.6 | – | 86.9 |
| InternVL2.5-8B (Chen et al., 2024a) | – | 39.5 | 64.4 | 19.7 | 2344.1 | 62.8 | 56.0 | – | **90.6** |
| *Reasoning Models* | | | | | | | | | |
| LLaVA-CoT-11B (Xu et al., 2024) | – | 20.3 | 54.8 | – | – | 60.3 | – | – | – |
| R1-Onevision-7B (Yang et al., 2025) | – | 46.4 | 64.1 | – | 2192.2 | 67.5 | – | 66.5 | 84.9 |
| OpenVLThinker-7B (Deng et al., 2025) | – | 47.9 | 70.2 | 25.3 | – | – | – | – | – |
| *Baselines with the Same Experiment Setting* | | | | | | | | | |
| Qwen2.5-VL-7B$^\star$ (Bai et al., 2025) | 46.6 | 48.2 | 68.2 | 21.3 | 2262.6 | 67.1 | 59.3 | 75.6 | 86.0 |
| w/ GRPO (Guo et al., 2025) | 54.9 | 50.6 | 68.1 | 22.5 | 2278.6 | 60.8 | 64.7 | 76.5 | 85.4 |
| w/ GRPO$^\dagger$ (Guo et al., 2025) | 53.1 | 51.3 | 69.5 | 18.8 | 2167.6 | 58.6 | 59.1 | 76.3 | 85.8 |
| *Our Methods* | | | | | | | | | |
| w/ D2D$_{loc}$ (ours) | 54.5 | 49.1 | 68.0 | 20.0 | 2358.4 | 65.3 | 66.0 | **77.3** | 87.1 |
| w/ D2I$_{loc}$ (ours) | **60.6** | 51.1 | 69.9 | 26.3 | 2267.1 | 67.9 | 61.1 | 76.6 | 86.8 |
| $\Delta_{base}$ | +14.0 | +2.9 | +1.7 | +5.0 | +4.5 | +0.8 | +1.8 | +1.0 | +0.8 |
| $\Delta_{grpo\dagger}$ | +7.5 | −0.2 | +0.4 | +7.5 | +99.5 | +9.3 | +2.0 | +0.3 | +1.0 |
| w/ D2D$_{jus}$ (ours) | 28.9 | 49.1 | 65.7 | 20.1 | 2351.1 | **69.7** | 64.4 | 77.0 | 86.9 |
| w/ D2I$_{jus}$ (ours) | **65.0** | **53.8** | 69.7 | 19.7 | 2299.0 | 66.9 | 61.4 | 76.4 | 85.8 |
| $\Delta_{base}$ | +18.4 | +5.6 | +1.5 | −1.6 | +36.4 | −0.2 | +2.1 | +0.8 | −0.2 |
| $\Delta_{grpo\dagger}$ | +11.9 | +2.5 | +0.2 | +0.9 | +131.4 | +8.3 | +2.3 | +0.1 | +0.0 |
| w/ D2D$_{par}$ (ours) | 52.1 | 47.6 | 62.6 | 17.6 | **2373.7** | 66.4 | 67.6 | 76.5 | **88.8** |
| w/ D2I$_{par}$ (ours) | 60.5 | 50.6 | **72.2** | 27.2 | 2219.8 | 65.3 | 61.6 | 76.1 | 86.3 |
| $\Delta_{base}$ | +13.9 | +2.4 | +4.0 | +5.9 | −42.8 | −1.8 | +2.3 | +0.5 | +0.3 |
| $\Delta_{grpo\dagger}$ | +7.4 | −0.7 | +2.7 | +8.4 | +52.2 | +6.7 | +2.5 | −0.2 | +0.5 |

and OpenVLThinker-7B (Deng et al., 2025)), and (4) baselines with the same experiment setting (*i.e.*, Qwen2.5-VL-7B (Bai et al., 2025) itself and with GRPO training (Guo et al., 2025)). We also compare D2I with D2D in our designed deliberate reasoning strategies.

# 6 EXPERIENTAL RESULTS AND ANALYSIS

We organize our experimental analysis around these following *Research Questions (RQs)*:

- **RQ1**: *How does quantitative performance of D2I trained with different deliberate reasoning strategies compare to competitive baselines?*
- **RQ2**: *Is D2I's improvement over D2D due to the model's lack of deductive reasoning capability?*
- **RQ3**: *How does the qualitative performance of D2I stand against competitive baselines?*
- **RQ4**: *What exactly does designed three atrategies, D2I and D2D influence in the models?*

## 6.1 MAIN RESULTS (**RQ1**)

Results of training on mathematical dataset are summarized in Table 1. D2I and D2D achieve strong performance across **both in-domain and out-of-domain benchmarks.** Specifically, D2I consistently outperforms the Qwen2.5-VL-7B base model and the GRPO baseline on nearly all benchmarks. For in-domain math evaluation, D2I improves performance by at least 13.9% over the base model and at least 7.4% over GRPO. On out-of-domain math benchmarks, D2I$_{loc}$ and D2I$_{par}$ both achieve gain of 1% to 8%, while they slightly drop on MathVerse. D2I$_{jus}$ achieves gain of 1% to 6%, while it slightly drop on MATH-Vision.

Even though D2I does not surpass GPT-4o on a few general benchmarks such as MATH-Vision and MMMU, it consistently outperforms other open-source general models and reasoning models. These results demonstrate the effectiveness of our training framework and the deliberate reasoning strategies in improving multimodal reasoning capabilities.

Table 2: Comparison of our method (RL-Only) with SFT-only and SFT-RL models. **Bold** highlights the best performance.

| Method | Out-of-domain (Math) | | | Out-of-domain (General) | | | | |
|---|---|---|---|---|---|---|---|---|
| | MathVerse$_{mini}$ | MathVista$_{mini}$ | MATH-Vision$_{full}$ | MME$_{sum}$ | MMVet$_{turbo}$ | MMMU$_{val}$ | SEED | POPE |
| *SFT-Only Models* | | | | | | | | |
| SFT$_{GEOQA}$ | 29.1 | 56.8 | 14.0 | 2196.6 | 46.2 | 61.6 | 77.1 | 87.9 |
| SFT$_{loc}$ | 43.2 | 59.6 | 21.8 | 2325.2 | 68.1 | 64.6 | 75.5 | 85.5 |
| SFT$_{par}$ | 27.7 | 51.0 | 14.8 | 1982.8 | 60.8 | 66.4 | 77.1 | 86.2 |
| *SFT-RL Models* | | | | | | | | |
| SFT−D2D$_{loc}$ | 46.0 | 56.7 | 17.0 | 2320.7 | 61.9 | 61.4 | 76.8 | 86.8 |
| SFT−D2I$_{loc}$ | 43.9 | 68.7 | 23.6 | 2318.7 | 65.8 | 64.0 | 75.9 | 81.2 |
| SFT−D2D$_{par}$ | 33.8 | 56.1 | 15.1 | 2353.2 | 46.8 | 66.3 | 75.2 | 88.3 |
| SFT−D2I$_{par}$ | 36.6 | 67.0 | 20.8 | 2302.5 | 60.6 | 65.8 | 76.9 | 86.4 |
| *RL-Only Models* | | | | | | | | |
| D2D$_{loc}$ (ours) | 49.1 | 68.0 | 20.0 | 2358.4 | 65.3 | 66.0 | **77.3** | 87.1 |
| D2I$_{loc}$ (ours) | 51.1 | 69.9 | 26.3 | 2267.1 | 67.9 | 61.1 | 76.6 | 86.8 |
| D2D$_{jus}$ (ours) | 49.1 | 65.7 | 20.1 | 2351.1 | **69.7** | 64.4 | 77.0 | 86.9 |
| D2I$_{jus}$ (ours) | **53.8** | 69.7 | 19.7 | 2299.0 | 66.9 | 61.4 | 76.4 | 85.8 |
| D2D$_{par}$ (ours) | 47.6 | 62.6 | 17.6 | **2373.7** | 66.4 | **67.6** | 76.5 | **88.8** |
| D2I$_{par}$ (ours) | 50.6 | **72.2** | **27.2** | 2219.8 | 65.3 | 61.6 | 76.1 | 86.3 |

We attribute this strong performance to two key factors: (1) our use of format-constrained training objectives that promote structured, interpretable reasoning patterns, and (2) free-form manner encourages the model to combine with more aggressive output strategies and a larger search space can significantly increase the likelihood of hitting the correct answer. Therefore, this leads to notable performance improvements on math benchmarks by avoiding generating low-quality deliberate reasoning outputs (*e.g.* box and parsing outputs). For general benchmarks, it may become overly divergent in its reasoning process, thereby missing the correct answers and resulting in a negative impact on overall performance. We also analyze the difference among the three designed strategies in Appendix D.

## 6.2 ABLATION STUDY *(RQ2)*

To investigate whether D2I's improvement over D2D is due to the model's lack of deductive reasoning capability, we compare against both SFT-only and SFT-RL variants. Details of training the compared model are shown in Appendix E. As shown in Table 2, our RL-only models consistently outperform both SFT-only and SFT-RL baselines across most benchmarks, regardless of whether the loc or par strategy is employed. Interestingly, we observe that within both the RL-only and SFT-RL setups, the D2I variant generally outperforms the corresponding D2D model. This suggests that the performance gap between D2D and D2I in RL-only training is not merely due to the inability of D2D to learn accurate box or parsing outputs; rather, it highlights the critical role of our reinforcement learning framework in enhancing model reasoning capabilities. Moreover, the SFT-RL variants of D2D do not outperform their RL-only counterparts, indicating that SFT is not the determining factor in driving performance within our ar-

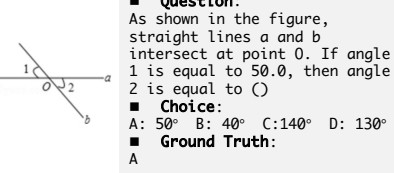

■ **Question**:
As shown in the figure, straight lines $a$ and $b$ intersect at point O. If angle 1 is equal to 50.0, then angle 2 is equal to ()
■ **Choice**:
A: 50° B: 40° C:140° D: 130°
■ **Ground Truth**:
A

**GRPO$^†$**

Since angle 1 and angle 2 are _adjacent angles_ . ... Therefore, the measure of angle 2 is 130 degrees. So, the correct answer is D

**D2I$_{loc}$**

Since angle 1 and angle 2 are _linear pairs, their sum is 180°_. ... Thus, the measure of angle 2 is 130°.The correct answer is: D

**D2I$_{jus}$**

Angle 1 and angle 2 are _vertical angles. Therefore, angle 1 is equal to angle 2._ Given that angle 1 is 50°, it follows that angle 2 is also 50°.Thus, the correct answer is A.

**D2I$_{par}$**

Since angle 1 and angle 2 are _adjacent angles_, ... Therefore, the measure of angle 2 is 130 degrees.The correct answer is D.

Figure 3: Response during inference of the same math question trained with different deliberate reasoning strategies.

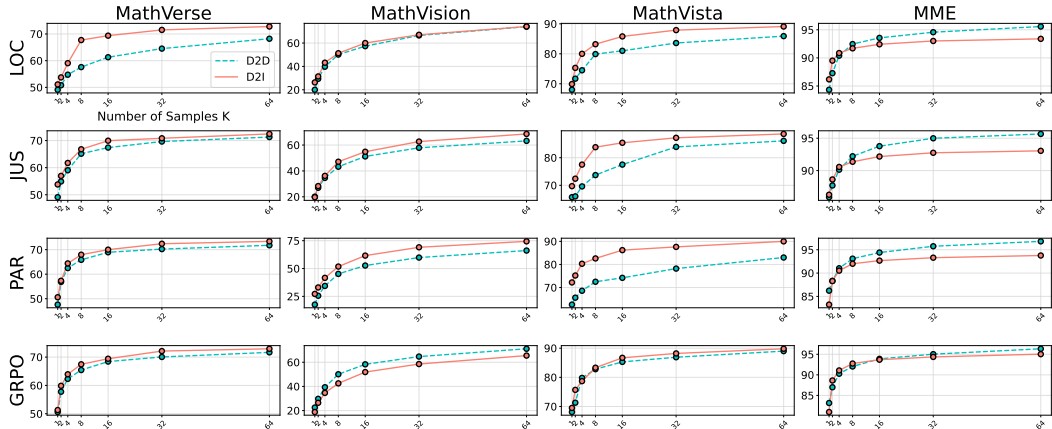

Figure 4: Pass@$k$ results on Math benchamrks (*i.e.* MathVerse, MathVistion and MathVista) and General benchmarks (*i.e.* MME).

chitecture. Additionally, we also include an SFT$_{GEOQA}$ trained on the same dataset as in our main results, which still underperforms compared to our RL-only models. This further underscores the superiority of our approach.

## 6.3 CASE STUDY *(RQ3)*

We analyze a representative failure case where GRPO, D2I$_{loc}$, and D2I$_{par}$ all produce incorrect answers, while D2I$_{jus}$ succeeds. As shown in Figure 3, notably, all four responses share a highly similar structure and format, demonstrating the same reasoning steps. However, only the D2I$_{jus}$ model correctly identifies a critical concept in its reasoning trace: the mention of *vertical angles*. This small but crucial difference allows D2I$_{jus}$ to reach the correct final answer, whereas the other models fail. This highlights an important observation: **Once the key visual perceptions are correctly understood, the model can form a complete and accurate reasoning chain, with no explicit output format or structured intermediate representation required**. It underscores that understanding and articulating the right concept, rather than strictly enforcing output formats, is often the deciding factor in successful reasoning. This case also illustrates the unique strength of D2I$_{jus}$ in aligning with the LLM's natural generation behavior.

## 6.4 MORE EXPLORATION *(RQ4)*

**Reasoning Upper Bound.** we analyze the Pass@$k$ (Chen et al., 2021) as shown in Figure 4. Across most benchmarks, D2I consistently outperforms D2D, particularly at smaller values of $k$. This indicates that D2I enables the model to generate more accurate responses in higher-ranked hypotheses, suggesting better guidance during inference and a higher upper bound. Moreover, we observe that in the GRPO baseline, the performance of D2D and D2I tends to converge as $k$ increases. In contrast, under our three deliberate reasoning strategies, the gap between D2D and D2I remains more pronounced. This indicates that deliberate reasoning strategies exert a stronger enhancing or suppressing effect on D2I, demonstrating their greater influence in shaping model behavior. It also highlights the effectiveness of deliberate reasoning strategies in improving sample efficiency, as D2I is more likely to hit the correct answer with fewer samples.

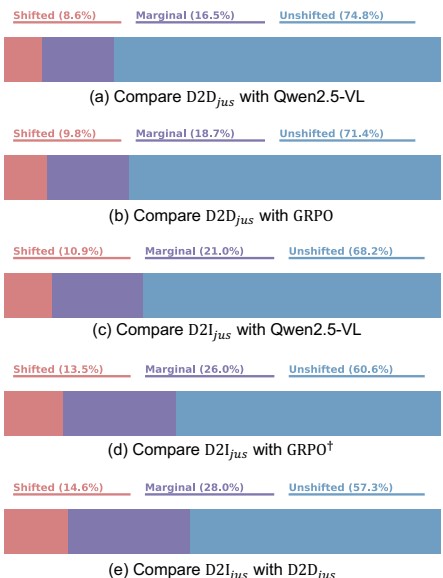

Figure 5: Results of token distribution shift for JUS strategy on MATH-Vision.

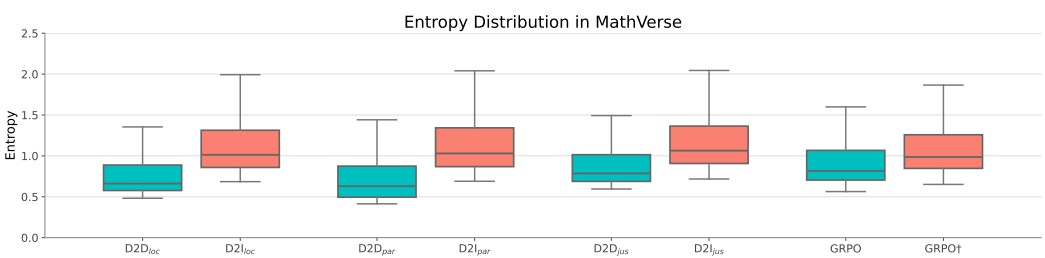

Figure 6: Entropy distribution results on MathVerse dataset.

**Entropy Distribution.** To further examine the exploration characteristics of our models, we conduct entropy-based analysis, measuring the output distribution entropy (Cheng et al., 2025a) for each model under different prompting conditions. Entropy serves as a proxy for how deterministic or exploratory a model's generation policy is: lower entropy implies more deterministic outputs, while higher entropy suggests a broader sampling space and increased diversity. We visualize the entropy distribution using box plots, comparing D2D and D2I across benchmarks as shown in Figure 6. The results reveal that D2I exhibits a broader and higher entropy distribution than D2D, confirming that D2I encourages more diverse and less deterministic generation patterns. We also show the entropy distribution on other benchmarks in Appendix F.

**Token Distribution Shift.** To investigate the behavioral differences between models during inference, we analyze how each model modifies its token-level output given the same input prompt (Lin et al., 2023). Specifically, we compare the response tokens of our deliberate reasoning strategy with those of baseline models, including GRPO-trained and base model variants. This comparison quantifies how aggressively each model reshapes its output distribution. As shown in Figure 5, shifted tokens refer to token positions where the selected token ranks differ by three or more between the two models, marginal tokens indicate a rank difference of one to two, and unshifted tokens represent positions where the two models select the same token. D2I models consistently introduce a higher proportion of shifted tokens relative to their D2D counterparts, indicating a more exploratory generation pattern. This pattern of token-level shift implies that D2I is not merely fine-tuning or adjusting local generation behavior but is actively restructuring the output space. The higher token shift ratio of D2I reflects its capacity to escape suboptimal local minima and explore alternative outputs, particularly beneficial in math benchmarks. We also show the token distribution shift results on other experimental settings in Appendix G and word cloud visualizations to more intuitively illustrate which specific tokens were altered in Appendix H.

# 7 CONCLUSION

In this work, we propose the Deliberate-to-Intuitive reasoning framework for improving MLLMs on complex visual reasoning tasks. Motivated by the gap between training-time supervision and inference-time behavior, D2I encourages models to engage in deeper reasoning during training through format-constrained, goal-driven reinforcement learning, while allowing flexible and intuitive generation at test time. We design three types of deliberate reasoning strategies to guide the model toward stronger semantic understanding and reasoning capability. Extensive experiments on both in-domain and out-of-domain benchmarks demonstrate that D2I consistently outperforms strong baselines across a wide range of reasoning challenges. Notably, D2I shows competitive or superior performance even compared to some proprietary models, despite relying only on lightweight rule-based rewards and no additional annotation. We believe that D2I offers a scalable, annotation-free strategy for enhancing the reasoning abilities of MLLMs. In future work, we plan to explore further applications to other multimodal domains such as science diagrams, instructional videos, or procedural planning.

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

## A  THE USE OF LARGE LANGUAGE MODELS

In the preparation of this manuscript, LLMs were used solely for the purpose of text polishing, including grammar correction and stylistic refinement. No content was generated or rewritten with the intention of altering the scientific meaning, originality, or conclusions of the work. All ideas, analyses, and results presented in this paper are entirely the authors' own.

## B  IMPLEMENTATION

We adopt Qwen2.5-VL-7B-Instruct (Bai et al., 2025) as our base model for all experiments. We train the model for 150 steps with the batch size of $128$, the learning rate of $1e-6$, the max response length of 1024 tokens, and the sampling temperature of $1$. The inference-time hyperparameters for any setting are kept consistent with those used during training. During training and evaluation, we utilize the prompt as shown in Appendix C. In D2I, we train the model with the designed deliberate reasoning strategies by prompts shown in Training Stage of Appendix C, and evaluate it with the intuitive reasoning style by prompts shown in Inference Stage of Appendix C. In D2D, during both training and evaluation, we use the same prompts with the designed deliberate reasoning strategies by prompts shown in Training Stage of Appendix C. All experiments are conducted on $8$ NVIDIA A100 GPUs, each with $80$GB of memory.

## C    PROMPTS

**Training Stage.** The prompt used in the LOC strategy is as follows:

> You are a helpful assistant. First think about the reasoning process for answering the question. As part of your reasoning, identify the crucial region of the image needed to answer the question. Before reaching your final conclusion in the reasoning, output the coordinate of this crucial region. Your response should follow this structure. The reasoning process, the coordinate of the crucial region for answering this question, and answer are enclosed within `<think> </think>`, `<box> </box>`, and `<answer> </answer>` tags, respectively, i.e., `<think>` Your step-by-step reasoning process... `<box>` (x1, y1), (x2, y2) `</box>` the following step-by-step reasoning process based on this region... `</think><answer>` answer here `</answer>`.

The prompt used in the JUS strategy is as follows:

> You are a helpful assistant. User asks questions, Assistant solves step by step. First think about reasoning process within `<think></think>` tags. In addition to the reasoning steps leading to your answer, identify the crucial part(s) of the image that are essential for solving the question. Describe why these parts are important and explain how they contribute to your reasoning within `<crucial></crucial>`. Final answer goes in `<answer> </answer>` tags. The overall response format is `<think>` reasoning here `</think>` `<crucial>` crucial part description here `</crucial>` `<answer>`final answer here `</answer>`.

The prompt used in the PAR strategy is as follows:

> You are a helpful assistant. You FIRST parse the image with structure language, and then think about the reasoning process as an internal monologue and finally provide the final answer. The parsing result MUST BE enclosed within `<parse> </parse>` tags. The reasoning process MUST BE enclosed within `<think> </think>` tags. The final answer MUST BE enclosed within `<answer> </answer>` tags. Note the parsing result is parsing the image with listing all the objects and their relationships using predicate format. The overall response format is `<parse>` parsing result here `</parse>` `<think>` reasoning process here `</think>` `<answer>` final answer here `</answer>`.

The prompt used in the GRPO training is as follows:

> A conversation between User and Assistant. The user asks a question, and the Assistant solves it. The assistant first thinks about the reasoning process in the mind and then provides the user with the answer. The reasoning process and answer are enclosed within `<think> </think>` and `<answer> </answer>` tags, respectively, i.e., `<think>` reasoning process here `</think>` `<answer>` answer here `</answer>`.

**Inference Stage.** The prompt used in the D2I framework is as follows:

> You are a helpful assistant. Please provide step-by-step reasoning process first, and then provide your final answer.

## D    COMPARISON OF DIFFERENT DELIBERATE REASONING STRATEGIES

As shown in Table 1, among the three deliberate reasoning strategies, we observe that their performance varies across benchmarks, with no single strategy consistently dominating the others. Instead, each strategy shows strengths on specific types of tasks, reflecting complementary reasoning capabilities. The JUS strategy tends to perform better on benchmarks such as MathVerse and MMVet, which

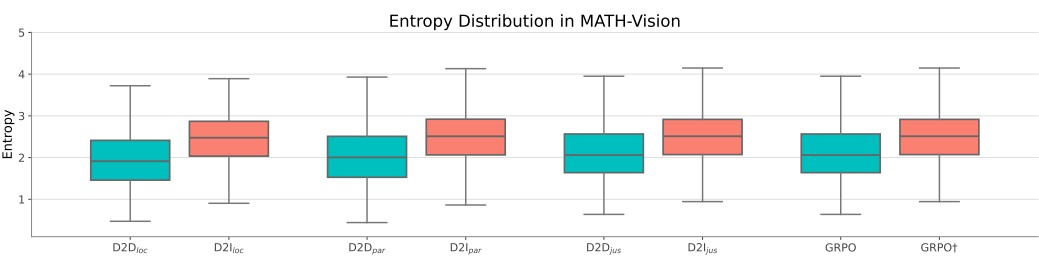

Figure 7: Entropy distribution results on MATH-Vision dataset.

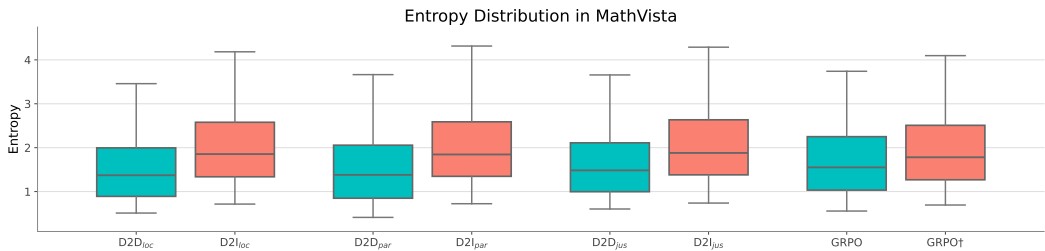

Figure 8: Entropy distribution results on MathVista dataset.

are explicitly designed to test whether MLLMs truly interpret math diagrams and demand expert-level multimodal understanding and reasoning. In both cases, **JUS's natural-language explanations help align the model's reasoning process** with the expectations of these benchmarks, making it easier to leverage visual cues in a text-friendly format. The LOC strategy shows strong results on MME and SEED, emphasizing accurate spatial grounding and fine-grained perception. By learning to identify bounding box coordinates of crucial regions, **LOC strategy enables the model to anchor its reasoning in the correct image region**, improving reliability in tasks sensitive to spatial detail. The PAR strategy performs well on MathVista, MATH-Vision, MMMU, and other benchmarks, which often involve complex visual layouts and structural relationships. For example, MathVista includes diverse visuals (*e.g.*, puzzle pieces, plots, and scientific diagrams), necessitating deep structural understanding. **The PAR strategy's emphasis on modeling global object structure helps the model form a coherent representation of the image**, which is critical for such benchmarks.

## E   TRAINING OF SFT-ONLY AND SFT-RL MODELS

Specifically, we construct two distinct SFT datasets to train SFT-only models as shown in Table 2. The *loc-sft* dataset focuses on predicting bounding box coordinates for crucial regions from mathematical images, enabling the model to learn accurate spatial localization. The SFT model trained on this dataset is referred to as $\text{SFT}_{loc}$. The *par-sft* dataset is designed for mathematical image parsing, aimed at teaching the model to generate correct parsing outputs; we denote the resulting model as $\text{SFT}_{par}$. Building upon these SFT models, we further update them using our proposed RL framework to form the SFT-RL models. By applying our method to $\text{SFT}_{loc}$, we obtain two models under corresponding deliberate reasoning strategy: $\text{SFT-D2D}_{loc}$ and $\text{SFT-D2I}_{loc}$. A similar procedure is applied to $\text{SFT}_{par}$, yielding $\text{SFT-D2D}_{par}$ and $\text{SFT-D2I}_{par}$.

## F   ENTROPY DISTRIBUTION

We show the entropy distribution on MATH-Vision in Figure 7 and on MathVista in Figure 8.

## G   TOKEN DISTRIBUTION SHIFT

We show the token distribution shift results in Figures 9, 10, 11, 12, 13, 14, 15, 16, 17, 18, 19 .

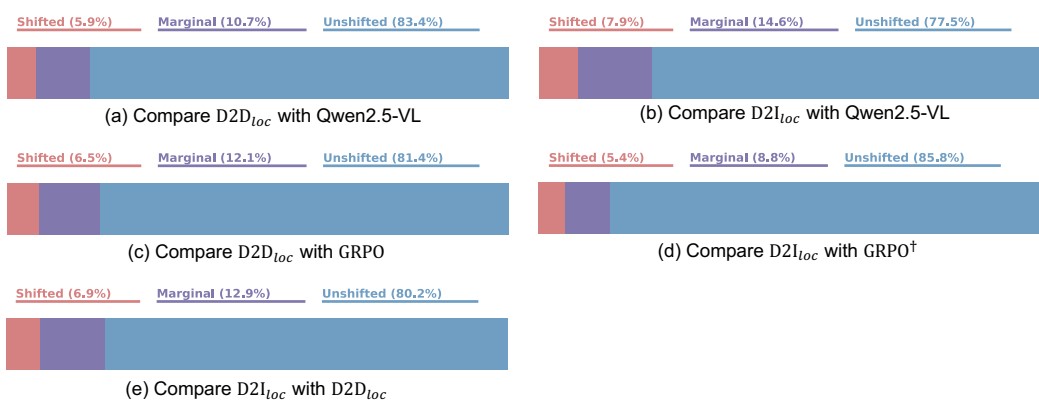

Figure 9: Token Distribution Shift of LOC on MathVerse dataset.

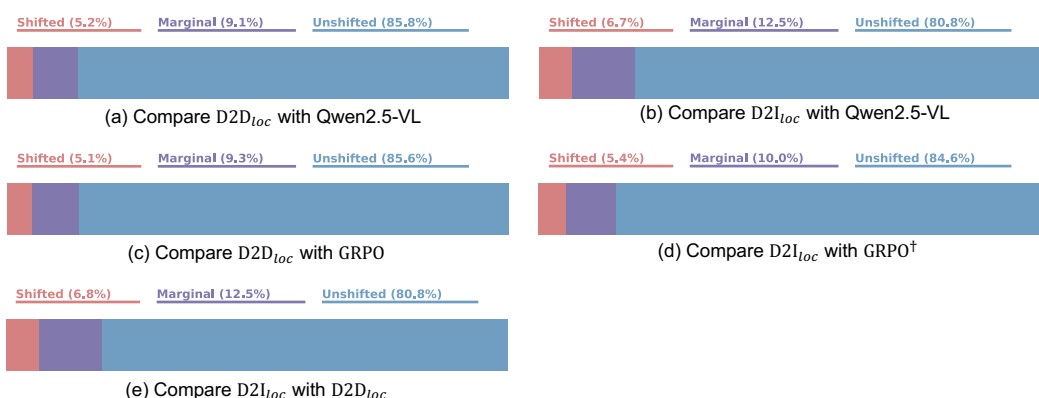

Figure 10: Token Distribution Shift of LOC on MathVista dataset.

## H WORD CLOUD VISUALIZATIONS

We also present word cloud visualizations to more intuitively illustrate which specific tokens were altered in MathVerse dataset. Results are shown in Figures 20, 21 and 22.

## I RESULTS ON NON-MATH TRAINING SET

To further verify whether the results are consistent on datasets beyond the mathematical domain, we constructed a mixed dataset in the document VQA field , which includes samples from DocVQA (Mathew et al., 2021), InfographicVQA (Mathew et al., 2022), ArxivQA (Li et al., 2024b), and TAT-DQA (Zhu et al., 2022; 2024). The training set contains 8,040 examples. For inference, we report the performance on the Doc-Mix test set as in-domain performance, which includes 720 examples. We also evaluate on the widely used out-of-domain benchmarks: MathVerse (Zhang et al., 2024b), MathVista (Lu et al., 2023), and MATH-Vision (Wang et al., 2024). Results are summarized in Table 3. Compared to other methods, the overall trend is not as strong as D2I trained on GEOQA-8K, since the model trained on document VQA data mostly achieved only the second-best performance. However, this is reasonable because we only used document VQA datasets and did not see any math-related samples during training. Nevertheless, our approach consistently outperforms GRPO and Qwen2.5-VL, demonstrating that our method maintains consistent effectiveness across different training data.

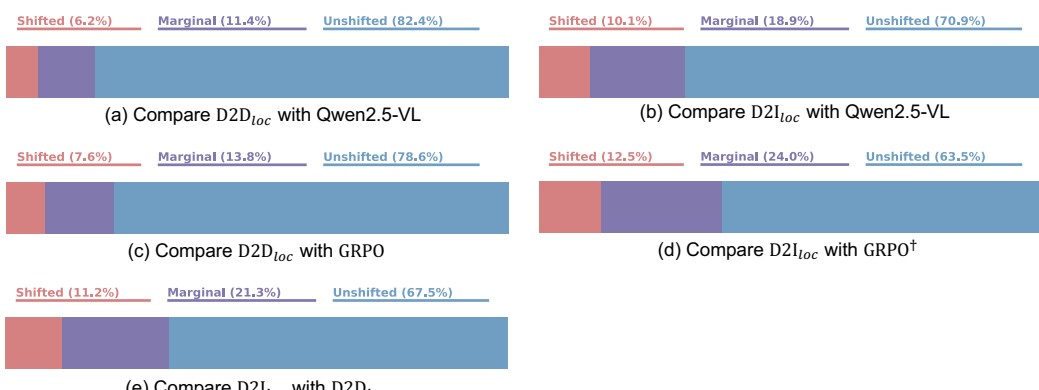

Figure 11: Token Distribution Shift of LOC on MATH-Vision dataset.

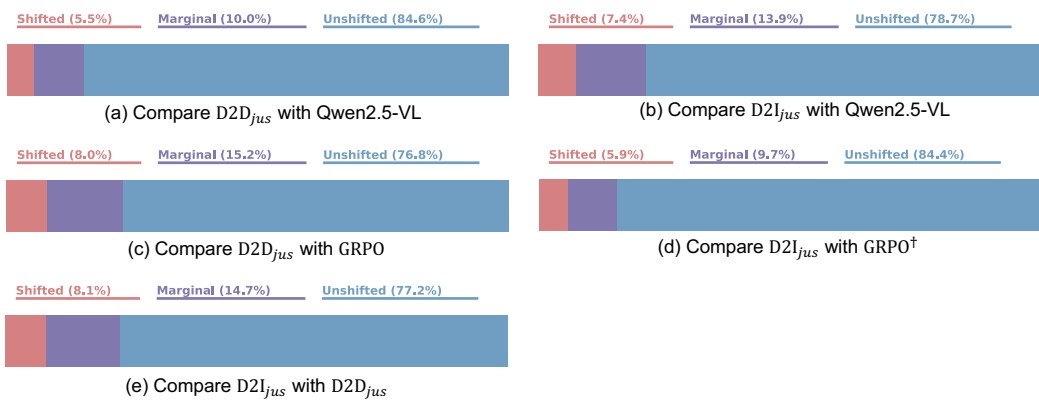

Figure 12: Token Distribution Shift of JUS on MathVerse dataset.

## J    DISCUSSION WITH OTHER REASONING METHODS

Recently, research on reasoning has gained significant traction. The following distinguished works have captured our attention, and we proceed here with a comparative analysis against our proposed framework:

1. Vision-R1 (Huang et al., 2025) introduced the core R1 paradigm that uses rule-based rewards to incentivize MLLMs to generate structured reasoning paths. It proved that format constraints can effectively enforce deliberate behavior and teach new skills without human content supervision. But the rigid requirement for structured output at inference time severely suppresses exploration and introduces brittleness, leading to suboptimal performance on OOD tasks compared to D2I.

2. R1-VL (Zhang et al., 2025) refined the R1 framework by introducing GRPO, focusing on step-wise relative quality feedback to optimize the policy. It provided the highly efficient and stable GRPO that D2I's deliberate training phase directly utilizes, significantly accelerating skill acquisition. But it is a coupled D2D model. The enhanced reasoning skill learned efficiently through GRPO is still constrained by the format requirement at inference. D2I unlocks this skill by removing the constraint.

3. Video-R1 (Feng et al., 2025) applied the R1 paradigm to video reasoning tasks, designing temporal-specific rule rewards to enhance multi-frame, deliberate processing in MLLMs. It validated the cross-modal generalizability of the R1 concept to complex sequence modalities like video. However, video reasoning is resource-intensive. Mandating structured D2D output exacerbates latency and inflexibility. D2I's principle of intuitive (concise) inference is even more crucial here to maintain efficiency and generalizability.

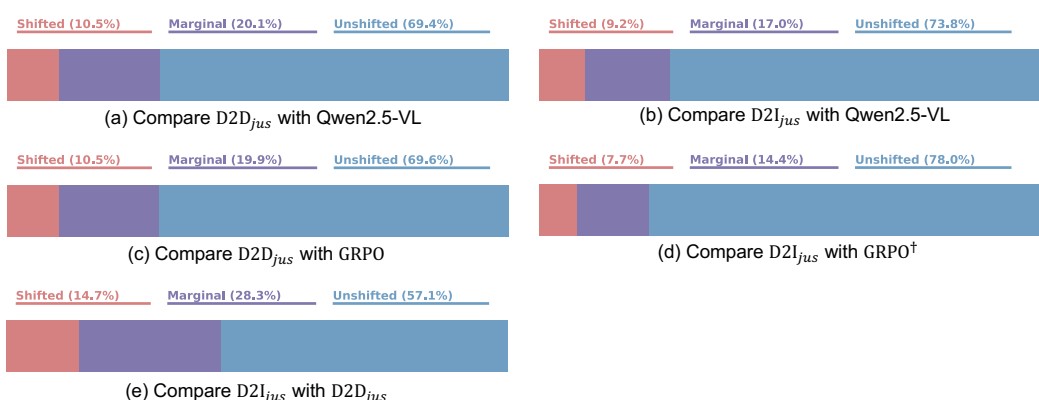

Figure 13: Token Distribution Shift of JUS on MathVista dataset.

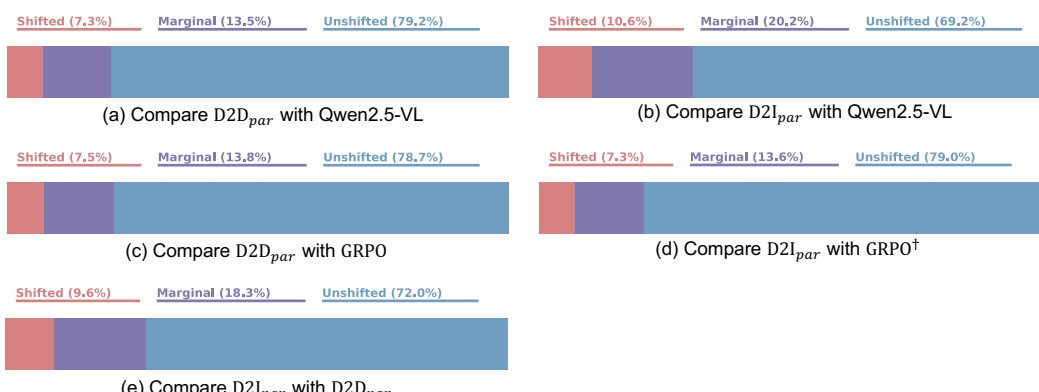

Figure 14: Token Distribution Shift of PAR on MathVerse dataset.

4. Seg-Zero (Liu et al., 2025) utilized R1-style reinforcement to guide and improve non-linguistic, structured visual output via a reasoning chain. It demonstrated that format reinforcement can drive the generation of structured outputs that are not pure text, linking thought chains to low-level visual perception tasks. But its D2D nature is necessary for the task. D2I's advantage is specific to reasoning tasks where the structure is an intermediate scaffold. D2I's success lies in strategically discarding the scaffold to maximize reasoning flexibility, a strategy Seg-Zero cannot adopt.

5. VLM-R1 (Shen et al., 2025) focused on creating a stable and generalizable R1-style VLM by addressing common instabilities in RL training, greatly improved the training stability and scalability of the R1 framework, providing a more robust foundation for any subsequent deliberate training. Despite improving generalization in training, it remains a coupled D2D model. D2I proves that true OOD robustness is achieved not merely by stabilizing the training, but by strategically decoupling the acquired skill from the constraining structure at inference time.

## K EFFECT OF MODEL SCALE

This part presents our study on cross-scale generalization, demonstrating how the performance of the D2I framework is affected by variations in model size. Results on Qwen2.5-VL-3B (Bai et al., 2025) are shown in Table 4. D2I significantly improves the performance of the smaller 3B model, achieving substantial relative gains. This confirms that the method is potent enough to boost even models with limited capacity.

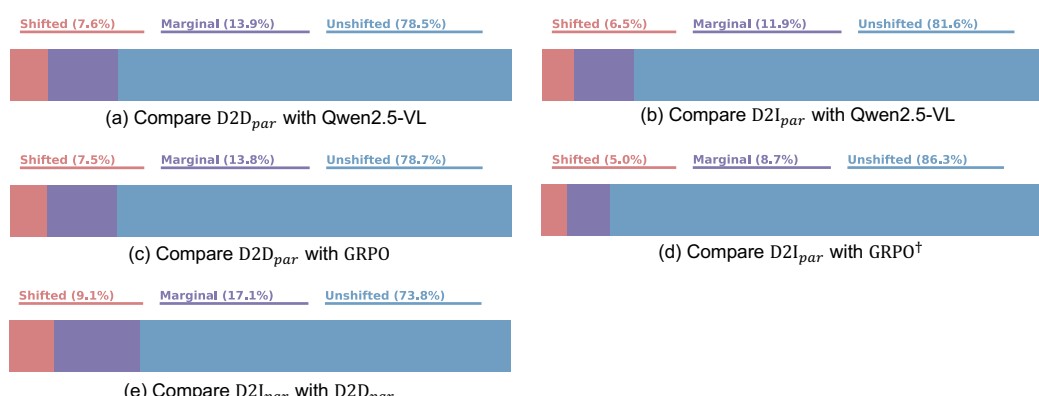

Figure 15: Token Distribution Shift of PAR on MathVista dataset.

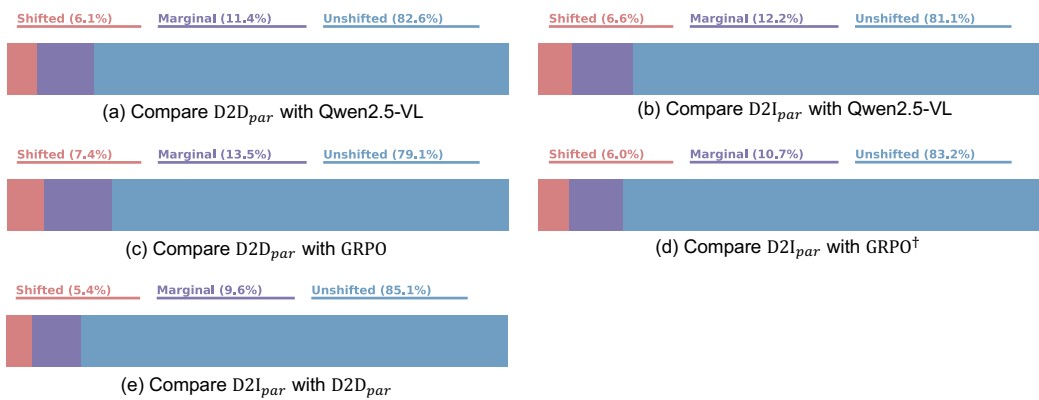

Figure 16: Token Distribution Shift of PAR on MATH-Vision dataset.

## L   EFFECT OF MLLM BACKBONE

We also study cross-architecture generalization, demonstrating how the performance of the D2I framework is affected by variations in different MLLM backbone. Results on InternVL2.5-8B (Chen et al., 2024c) are shown in Table 5. Applying D2I training to the InternVL2.5-8B architecture, which utilizes a different visual encoder and MLLM family, resulted in similar performance uplifts on key OOD reasoning benchmarks. This strongly validates that D2I is a generalizable training paradigm, not a tuning trick specific to the Qwen family.

## M   MORE CASE ANALYSIS

Analyzing D2D failures is also crucial. We found several cases shown in Fig 23, particularly in the LOC strategy, where the model correctly identified a reasonable crucial region with the bounding box, but nonetheless proceeded to reach the wrong final answer in its reasoning chain. We attribute this persistent failure to these factors inherent in the D2D paradigm:

1. Generating a coordinate (LOC) is an isolated act that directs the model's attention to a region. However, merely attending to a region does not guarantee that the model will successfully interpret its semantic meaning and apply it correctly within the subsequent reasoning steps. The ability to truly enhance reasoning relies on the fusion of visual analysis with high-level semantic reasoning—a complex, non-explicit ability. In the D2D paradigm, the explicit generation of the coordinate often becomes a detour or a symbolic checkpoint that is treated separately from the actual language inference, failing to fully integrate the visual insight into the core reasoning policy.

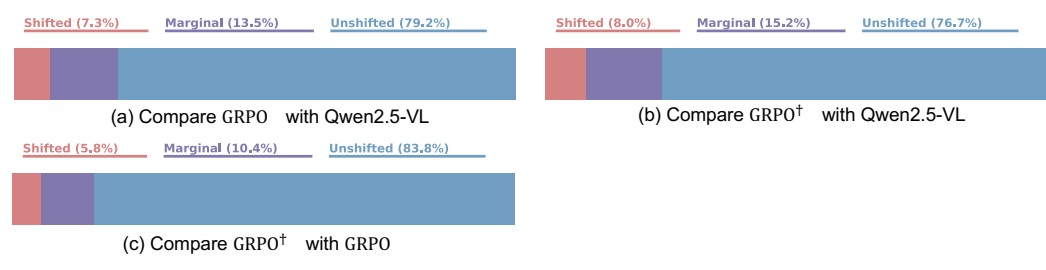

Figure 17: Token Distribution Shift of GRPO on MathVerse dataset.

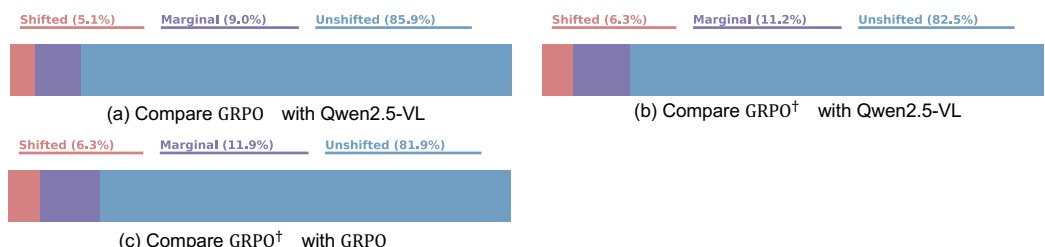

Figure 18: Token Distribution Shift of GRPO on MathVista dataset.

2. In the D2D paradigm, the model is conditioned on its own generated deliberate steps. If the model generates a plausible but slightly incorrect parsing or bounding box (which happens often, as these are hard tasks), the subsequent reasoning is hard-constrained to follow this incorrect premise. The model essentially talks itself into a wrong answer.

3. The D2D paradigm, by mandating the specific structural output (e.g., `<box>...</box>` or `<parse>...</parse>`) at test time, severely constrains the model's capacity for autonomous exploration. The model is rigidly bound to the self-generated artifact it just produced. When the model generates a plausible but ultimately suboptimal artifact (like a correct bounding box but the wrong interpretation of that box), the D2D constraint prevents the model from exploring alternative reasoning paths that could ignore the misleading artifact and find the correct solution. On the contrary, by removing the explicit deliberate tokens at test time, D2I removes this hard conditioning. The model utilizes the implicit capability (visual understanding) but is free to explore a reasoning path that isn't locked into a specific, potentially erroneous pre-generated structure.

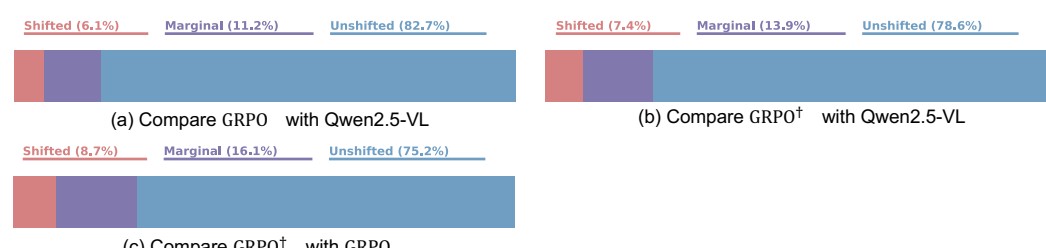

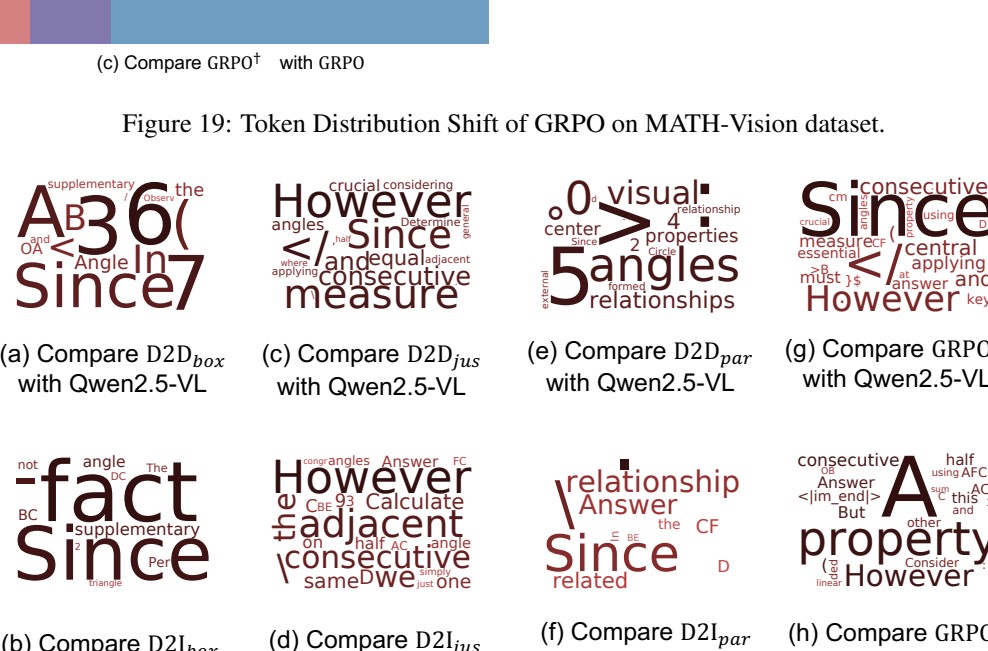

Figure 19: Token Distribution Shift of GRPO on MATH-Vision dataset.

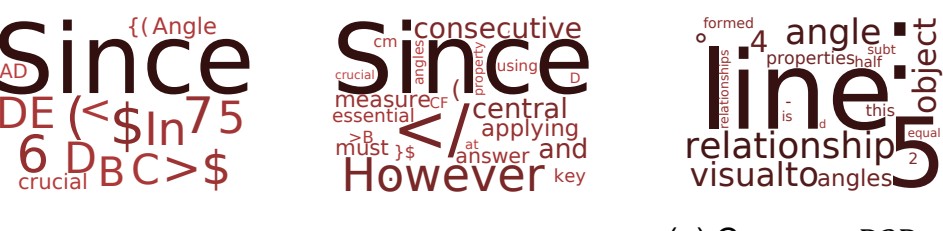

(a) Compare D2D$_{box}$ with Qwen2.5-VL

(b) Compare D2I$_{box}$ with Qwen2.5-VL

(c) Compare D2D$_{jus}$ with Qwen2.5-VL

(d) Compare D2I$_{jus}$ with Qwen2.5-VL

(e) Compare D2D$_{par}$ with Qwen2.5-VL

(f) Compare D2I$_{par}$ with Qwen2.5-VL

(g) Compare GRPO with Qwen2.5-VL

(h) Compare GRPO$^{\dagger}$ with Qwen2.5-VL

Figure 20: Word cloud visualizations on MathVerse.

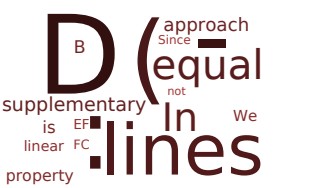

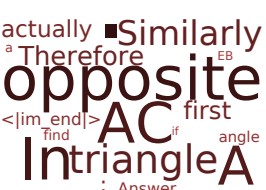

(a) Compare D2D$_{box}$ with GRPO

(b) Compare D2I$_{box}$ with GRPO$^{\dagger}$

(c) Compare D2D$_{jus}$ with GRPO

(d) Compare D2I$_{jus}$ with GRPO$^{\dagger}$

(e) Compare D2D$_{par}$ with GRPO

(f) Compare D2I$_{par}$ with GRPO$^{\dagger}$

Figure 21: Word cloud visualizations on MathVerse.

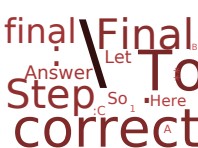 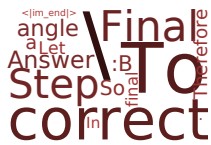 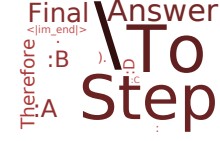 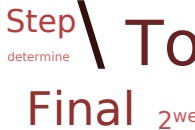

(a) Compare $D2I_{box}$ with $D2D_{box}$    (b) Compare $D2I_{jus}$ with $D2D_{jus}$    (c) Compare $D2I_{par}$ with $D2D_{par}$    (d) Compare $GRPO^{\dagger}$ with GRPO

Figure 22: Word cloud visualizations on MathVerse.

Table 3: Main results of our proposed D2I models trained on self-constructed non-math dataset. The **bold** denotes the best performance, while the underline denotes the second best one. In the Qwen w/ GRPO, we evaluate the model with the deliberate reasoning, while in w/ GRPO$^{\dagger}$ we evaluate with the intuitive reasoning. $\Delta_{base}$ represents the improvement of D2I over Qwen2.5-VL-7B, while $\Delta_{grpo\dagger}$ represents the improvement over Qwen2.5-VL-7B w/ GRPO$^{\dagger}$. $\star$ denotes our re-implementation.

| Method | Out-of-domain (Math) | | |
|---|---|---|---|
| | MathVerse (mini) | MathVista (mini) | MATH-Vision (full) |
| *Closed-Source General Models* | | | |
| GPT-4V (OpenAI, 2023) | 39.4 | 58.1 | 22.7 |
| GPT-4o (Hurst et al., 2024) | **50.2** | 63.8 | **30.3** |
| *Open-Source General Models* | | | |
| Qwen2-VL-7B (Bai et al., 2023) | 31.9 | 58.2 | 16.3 |
| InternVL2-8B (Chen et al., 2024b) | 37.0 | 58.3 | 18.4 |
| InternVL2.5-8B (Chen et al., 2024a) | 39.5 | 64.4 | 19.7 |
| *Reasoning Models* | | | |
| LLaVA-CoT-11B (Xu et al., 2024) | 20.3 | 54.8 | – |
| R1-Onevision-7B (Yang et al., 2025) | 46.4 | 64.1 | – |
| OpenVLThinker-7B (Deng et al., 2025) | 47.9 | **70.2** | 25.3 |
| *Baselines with the Same Experiment Setting* | | | |
| Qwen2.5-VL-7B$^{\star}$ (Bai et al., 2025) | 48.2 | 68.2 | 21.3 |
|   w/ GRPO (Guo et al., 2025) | 47.4 | 67.9 | 23.0 |
|   w/ GRPO$^{\dagger}$ (Guo et al., 2025) | 43.3 | 65.4 | 19.1 |
| *Our Methods* | | | |
|   w/ $D2D_{loc}$ (ours) | 42.2 | 64.8 | 20.3 |
|   w/ $D2I_{loc}$ (ours) | 48.8 | 68.9 | 22.2 |
| $\Delta_{base}$ | +0.6 | +0.7 | +0.9 |
| $\Delta_{grpo\dagger}$ | +5.5 | +3.5 | +3.1 |
|   w/ $D2D_{jus}$ (ours) | 45.8 | 64.7 | 19.7 |
|   w/ $D2I_{jus}$ (ours) | 48.6 | 70.1 | 21.3 |
| $\Delta_{base}$ | +0.4 | +1.9 | +0.0 |
| $\Delta_{grpo\dagger}$ | +5.3 | +4.7 | +2.2 |

Table 4: Main results of our proposed GEOQA-8K-trained D2I method on both in-domain and out-of-domain test sets. $\Delta_{base}$ represents the improvement of D2I over Qwen2.5-VL-3B, while $\Delta_{grpo\dagger}$ represents the improvement over Qwen2.5-VL-3B w/ GRPO$^\dagger$. $\star$ denotes our re-implementation. All the experiments are conducted on Qwen2.5-VL-3B.

| Method | In-domain | Out-of-domain (Math) | | | Out-of-domain (General) | | | | |
| | GEOQA-8K | MathVerse (mini) | MathVista (mini) | MATH-Vision (full) | MME (sum) | MMVet (turbo) | MMMU (val) | SEED | POPE |
| --- | --- | --- | --- | --- | --- | --- | --- | --- | --- |
| *Baselines with the Same Experiment Setting* | | | | | | | | | |
| Qwen2.5-VL-3B$^\star$ | 35.3 | 38.2 | 61.9 | 17.9 | 2102.8 | 60.2 | 52.9 | 62.4 | 71.0 |
| w/ GRPO | 46.1 | 37.1 | 56.3 | 17.7 | 2154.6 | 57.0 | 53.1 | 60.9 | 69.8 |
| w/ GRPO$^\dagger$ | 46.4 | 37.5 | 57.2 | 16.2 | 2132.5 | 57.2 | 52.4 | 61.2 | 70.3 |
| *Our Methods* | | | | | | | | | |
| w/ D2D$_{loc}$ (ours) | 48.4 | 42.4 | 61.0 | 19.4 | 2170.2 | **61.7** | 54.0 | 60.8 | 70.0 |
| w/ D2I$_{loc}$ (ours) | 50.2 | 43.1 | 64.2 | **20.9** | 2146.3 | 59.9 | 52.6 | 61.9 | 70.4 |
| w/ D2D$_{jus}$ (ours) | 40.3 | 40.2 | 60.3 | 18.5 | 2140.8 | 61.3 | 53.8 | 60.2 | 69.5 |
| w/ D2I$_{jus}$ (ours) | **52.1** | **43.7** | 63.2 | 17.3 | 2129.0 | 55.7 | 53.1 | 61.0 | 67.6 |
| w/ D2D$_{par}$ (ours) | 50.0 | 40.3 | 60.2 | 16.9 | **2231.5** | 60.8 | **56.2** | 64.9 | **73.5** |
| w/ D2I$_{par}$ (ours) | **52.1** | 42.3 | 62.7 | 19.1 | 2174.1 | 57.2 | 51.3 | **66.3** | 72.1 |

Table 5: Main results of our proposed GEOQA-8K-trained D2I method on both in-domain and out-of-domain test sets. $\Delta_{base}$ represents the improvement of D2I over InternVL2.5-8B, while $\Delta_{grpo\dagger}$ represents the improvement over InternVL2.5-8B w/ GRPO$^\dagger$. $\star$ denotes our re-implementation. All the experiments are conducted on InternVL2.5-8B.

| Method | In-domain | Out-of-domain (Math) | | | Out-of-domain (General) | | | | |
| | GEOQA-8K | MathVerse (mini) | MathVista (mini) | MATH-Vision (full) | MME (sum) | MMVet (turbo) | MMMU (val) | SEED | POPE |
| --- | --- | --- | --- | --- | --- | --- | --- | --- | --- |
| *Baselines with the Same Experiment Setting* | | | | | | | | | |
| InternVL2.5-8B$^\star$ | 40.8 | 37.2 | 61.0 | 19.2 | 2298.1 | 61.9 | 54.2 | 64.2 | 84.8 |
| w/ GRPO | 52.7 | 41.8 | 67.1 | 20.8 | 2319.3 | 64.7 | 59.9 | 67.1 | 88.2 |
| w/ GRPO$^\dagger$ | 57.4 | 45.6 | 67.8 | 19.3 | 2322.8 | 60.4 | 60.1 | 66.4 | 85.9 |
| *Our Methods* | | | | | | | | | |
| w/ D2D$_{loc}$ (ours) | 54.2 | 44.9 | 69.2 | 21.0 | **2358.2** | 62.8 | 63.7 | 69.1 | 89.2 |
| w/ D2I$_{loc}$ (ours) | 59.2 | **47.1** | **69.8** | 22.4 | 2273.6 | 63.3 | **64.0** | 68.7 | 88.5 |
| w/ D2D$_{jus}$ (ours) | 53.8 | 44.2 | 67.5 | 17.9 | 2301.6 | **65.3** | 62.3 | **69.8** | **89.3** |
| w/ D2I$_{jus}$ (ours) | 50.3 | 45.2 | 64.7 | 21.4 | 2326.7 | 61.0 | 61.8 | 66.2 | 82.0 |
| w/ D2D$_{par}$ (ours) | 54.0 | 42.9 | 68.1 | 20.4 | 2340.9 | 62.7 | 60.9 | 67.5 | 88.7 |
| w/ D2I$_{par}$ (ours) | 57.7 | 45.8 | 69.3 | **23.9** | 2306.3 | 60.4 | 60.4 | 67.9 | 89.0 |

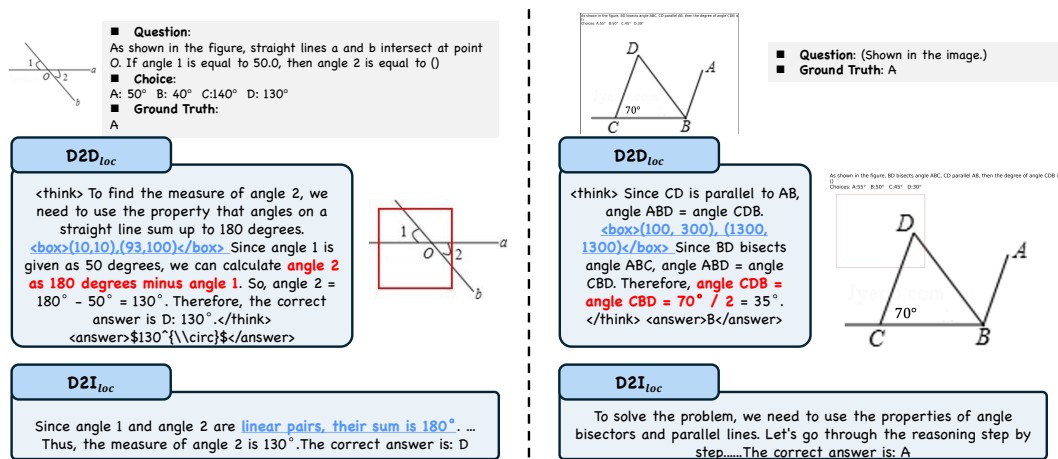

Figure 23: Failed cases on D2D paradigm. The red box in the image next to the D2D$_{loc}$ response is drawn based on the coordinates provided in the response, serving as a visual representation of the area the model focused on within the `<box></box>`.

