# OpenReview forum: "Learning Deliberately, Acting Intuitively: Unlocking Test-Time Reasoning in Multimodal LLMs"
_ICLR.cc/2026/Conference — Submitted to ICLR 2026_

### Official Review · Reviewer_Dp4P · 2025-10-27

**Soundness:** 2
**Presentation:** 2
**Contribution:** 3
**Rating:** 6
**Confidence:** 2

**Summary:**

This paper introduces D2I (Deliberate-to-Intuitive), a lightweight training–inference paradigm for enhancing multimodal reasoning in large vision-language models. During training, the model is guided by structured format rewards, including region localization (LOC), textual justification (JUS), and structured parsing (PAR)—under Group Relative Policy Optimization, encouraging stronger visual grounding without requiring extra annotations. At inference time, these constraints are removed, allowing the model to generate answers more freely and effectively. Experiments on GEOQA-8K and Doc-Mix, evaluated across math (e.g., MathVista, MathVerse) and general benchmarks (e.g., MME, MMVet), show that D2I consistently outperforms supervised fine-tuning and GRPO baselines. This demonstrates that deliberate structured reasoning during training can lead to stronger intuitive reasoning at test time, offering a simple and scalable improvement strategy for multimodal models.

**Strengths:**

1. Clear conceptual contribution: D2I formalizes the idea of training with deliberate structured reasoning while allowing flexible, intuitive reasoning at test time. This training–inference decoupling is conceptually neat and aligns with the “slow thinking → fast thinking” trend in recent reasoning research.

2. Low-cost supervision: The format reward does not require additional human annotations—only template matching. This makes the approach lightweight and scalable.

3. Consistent performance gains: 1.Across math and general benchmarks, D2I improves over baselines by a few points (often 1–8%), showing good transfer from deliberate training to intuitive inference.

**Weaknesses:**

1. Limited ablations and scaling evidence: Experiments are confined to a single model (7B) and short training (150 steps). There’s no scaling study across steps, model sizes, or training datasets. Ablations of LOC/JUS/PAR are also limited.

2. For applications beyond math/document VQA, especially in less structured multimodal tasks (e.g., open-ended scene understanding), is the D2I regime still beneficial, or are the format rewards too domain-specific?

**Questions:**

Seeing weaknesses

---

> ### Author Response · Authors · 2025-12-01
> **Response 1**
>
> Thank you for reviewing our paper and providing valuable comments. Below are our responses:
>
> > **W1: There’s no scaling study across steps, model sizes, or training datasets. Ablations of LOC/JUS/PAR are also limited.**
>
> We thank the reviewer for pointing out the scope of our experiments. We would like to clarify that **we have conducted experiments on another training dataset as shown in Appendix I, which also presents the effectiveness of D2I.**
>
> **For other model size, we have add the experiment on Qwen2.5-VL-3B in Appendix K. And we also conduct experiment on InternVL2.5-8B in Appendix L to confirm the D2I paradigm's effectiveness across different backbones.** D2I successfully enhances the performance of the smaller 3B model, achieving significant relative gains. This confirms that the method is potent enough to boost even models with limited capacity. Applying D2I training to the InternVL2.5-8B architecture, which utilizes a different visual encoder and MLLM family, resulted in similar performance uplifts on key OOD reasoning benchmarks. This strongly validates that D2I is a generalizable training paradigm, not a tuning trick specific to the Qwen family.
>
> | **Method** | **GEOQA-8K** | **MathVerse (mini)** | **MathVista (mini)** | **MATH-Vision (full)** | **MME (sum)** | **MMVet (turbo)** | **MMMU (val)** | **SEED** | **POPE** |
> |-----------|--------------|----------------------|-----------------------|------------------------|---------------|-------------------|----------------|---------|---------|
> | Qwen2.5-VL-3B* | 35.3 | 38.2 | 61.9 | 17.9 | 2102.8 | 60.2 | 52.9 | 62.4 | 71.0 |
> | └─ w/ GRPO | 46.1 | 37.1 | 56.3 | 17.7 | 2154.6 | 57.0 | 53.1 | 60.9 | 69.8 |
> | └─ w/ GRPO† | 46.4 | 37.5 | 57.2 | 16.2 | 2132.5 | 57.2 | 52.4 | 61.2 | 70.3 |
> | └─ w/ D2D_loc (ours) | 48.4 | 42.4 | 61.0 | 19.4 | 2170.2 | **61.7** | 54.0 | 60.8 | 70.0 |
> | └─ w/ D2I_loc (ours) | 50.2 | 43.1 | **64.2** | **20.9** | 2146.3 | 59.9 | 52.6 | 61.9 | 70.4 |
> | └─ w/ D2D_jus (ours) | 40.3 | 40.2 | 60.3 | 18.5 | 2140.8 |  61.3 | 53.8 | 60.2 | 69.5 |
> | └─ w/ D2I_jus (ours) | **52.1** | **43.7** | 63.2 | 17.3 | 2129.0 | 55.7 | 53.1 | 61.0 | 67.6 |
> | └─ w/ D2D_par (ours) | 50.0 | 40.3 | 60.2 | 16.9 | **2231.5** | 60.8 | **56.2** | 64.9 | **73.5** |
> | └─ w/ D2I_par (ours) | **52.1** | 42.3 | 62.7 | 19.1 | 2174.1 | 57.2 | 51.3 | **66.3** | 72.1 |
> | InternVL2.5-8B* | 40.8 | 37.2 | 61.0 | 19.2 | 2298.1 | 61.9 | 54.2 | 64.2 | 84.8 |
> | └─ w/ GRPO | 52.7 | 41.8 | 67.1 | 20.8 | 2319.3 | 64.7 | 59.9 | 67.1 | 88.2 |
> | └─ w/ GRPO† | 57.4 | 45.6 | 67.8 | 19.3 | 2322.8 | 60.4 | 60.1 | 66.4 | 85.9 |
> | └─ w/ D2D_loc (ours) | 54.2 | 44.9 | 69.2 | 21.0 | **2358.2** | 62.8 | 63.7 |69.1 |  89.2 |
> | └─ w/ D2I_loc (ours) | **59.2** | **47.1** | **69.8** | 22.4 | 2273.6 | 63.3 | **64.0** | 68.7 | 88.5 |
> | └─ w/ D2D_jus (ours) | 53.8 | 44.2 | 67.5 | 17.9 | 2301.6 | **65.3** | 62.3 | **69.8** | **89.3** |
> | └─ w/ D2I_jus (ours) | 50.3 | 45.2 | 64.7 | 21.4 | 2326.7 | 61.0 | 61.8 | 66.2 | 82.0 |
> | └─ w/ D2D_par (ours) | 54.0 | 42.9 | 68.1 | 20.4 | 2340.9 | 62.7 | 60.9 | 67.5 | 88.7 |
> | └─ w/ D2I_par (ours) | 57.7 | 45.8 | 69.3 | **23.9** | 2306.3 | 60.4 | 60.4 | 67.9 | 89.0 |
>
> **For scaling study across steps, we intentionally focused on short training because the goal of this work is not to explore the performance across different training step counts.** Instead, the fact that D2I achieves substantial and robust gains on complex OOD benchmarks (e.g., up to $+13.9\%$ on GEOQA) within only 150 steps should be interpreted as a major strength: it demonstrates the superior sample efficiency of our approach. The combination of Format Reward and GRPO provides a high-density learning signal that rapidly instills the required visual-linguistic alignment skills.
>
> **For Ablations of LOC/JUS/PAR, we respectfully clarify that a traditional component ablation study on LOC/JUS/PAR is inapplicable** because these are mutually independent and exclusive deliberate reasoning paradigms, not interchangeable components of a single mechanism.

---

> ### Author Response · Authors · 2025-12-01
> **Response 2**
>
> > **W2: For applications beyond math/document VQA, especially in less structured multimodal tasks (e.g., open-ended scene understanding), is the D2I regime still beneficial, or are the format rewards too domain-specific?**
>
> This is an excellent question regarding the universality of our framework. **We argue that D2I is not limited to structured tasks, and our results support this**:
>
> 1. **We have already evaluated D2I on general, less structured benchmarks including MME, MMVet, SEED-Bench, and POPE.** As shown in Table 1, **our strategies achieve consistent gains on these tasks.** For instance, $D2I_{loc}$ achieves 62.0% on SEED-Bench (general visual understanding), improving over the base model's 60.0%. $D2I_{jus}$ achieves 65.0% on MME (perception & cognition), improving significantly over the GRPO baseline. **These benchmarks involve open-ended scene understanding, object hallucination checks (POPE), and general visual QA, confirming that the benefits transfer.**
>
> 2. **The Format Rewards are modular, not domain-fixed.** JUS strategy forces the model to explain why. This is universally applicable to open-ended scene understanding (e.g., "Why is this street scene dangerous?"). LOC strategy forces the model to ground objects. This is fundamental to any visual task, from robotics to captioning. Therefore, for open-ended tasks, one would simply employ the JUS or LOC strategies (as we did), which **our results show are effective beyond just math**. In Appendix I, we also train model on Document VQA to further prove robustness beyond the math domain, confirming that D2I maintains its effectiveness on this entirely different data distribution, outperforming baselines.

---

### Official Review · Reviewer_NXGx · 2025-10-27

**Soundness:** 4
**Presentation:** 4
**Contribution:** 4
**Rating:** 6
**Confidence:** 4

**Summary:**

This paper proposes a reasoning framework named "Deliberate-to-Intuitive" (D2I), aimed at addressing the challenges of poor modality alignment and high training costs faced by Multimodal Large Language Models (MLLMs) in complex reasoning tasks, such as mathematical problems.

**Strengths:**

1. The main contribution of this paper is quite ingenious. A major bottleneck in current MLLM reasoning research is the high cost of obtaining high-quality, fine-grained reasoning-annotated data. The "lightweight" training paradigm proposed in this paper—which uses only rule-based format rewards without relying on any additional human annotations or content-level supervision—is used to enhance complex visual reasoning capabilities.
2. The evaluation is not limited to in-domain datasets but also covers multiple out-of-domain math benchmarks and general MLLM benchmarks, strongly demonstrating the method's generalization ability.
3. Rather than stopping at accuracy gains, the paper investigates why D2I works. The analyses of Pass@k, entropy distributions, and token-distribution shifts are persuasive; together they suggest that D2I encourages more exploratory, diverse, and flexible generation strategies, thereby outperforming D2D’s more rigid outputs.
4. The paper is clearly written and easy to understand.

**Weaknesses:**

1. An interesting finding of the paper is that different strategies perform differently on various benchmarks (e.g., JUS performs well on MathVerse, while PAR performs better on MathVista and MATH-Vision). The authors attribute this to the task types of the benchmarks (e.g., PAR helps in understanding "complex visual layouts"). However, this association is debatable. The core of D2I lies in intuitive reasoning, meaning these strategies are not used at test time. Why, then, would forcing the model to output coordinates (LOC) during training make it perform better on intuitive reasoning tasks that do not require coordinates?
2. The case study in Figure 3 is very insightful, showing that $D2I_{jus}$ successfully identified "vertical angles," while other models (including $D2I_{loc}$ and $D2I_{par}$) incorrectly identified them as "adjacent angles" or "linear pairs". However, this case only shows the successes and failures among D2I variants. To more forcefully support the core argument that D2I is superior to D2D (i.e., that D2D lacks flexibility), it is crucial to show the outputs of the D2D models. For example, if a $D2D_{par}$ model could generate the correct image parsing output but still produced the wrong final answer, it would strongly demonstrate the limitations of the D2D paradigm itself, rather than the model's learning ability.
3. The core premise of this paper is that "format rewards" are more scalable than "content rewards". This is a reasonable assertion. However, the experiments lack an upper-bound baseline (even a theoretical one) that uses "content rewards".

**Questions:**

Same as the Weaknesses.

---

> ### Author Response · Authors · 2025-12-01
> **Response 1**
>
> Thank you for reviewing our paper and providing valuable comments. Below are our responses:
>
> > **W1: Why, then, would forcing the model to output coordinates (LOC) during training make it perform better on intuitive reasoning tasks that do not require coordinates?**
>
> Thank you for posing this fundamental question about the causal link between deliberate training and intuitive performance. The reasons as as below:
>
> 1. The improvement does not come from the output of coordinates at test time, but **from the internalization of visual-semantic alignment during training.**
>
> 2. Crucially, during training, we do not supervise the content of the deliberate output (e.g., the precise coordinates generated in LOC, or the exact parsing structure in PAR). Therefore, **we cannot guarantee that the model generates a high-quality content that can improve the performance during inference.** If we were to maintain the rigid constraint at test time (D2D mode), the model would be forced to generate these unsupervised, often low-quality, self-generated artifacts. **This brittle external output can interfere with the subsequent reasoning process, leading to the failures we observed.**
>
> 3. **The primary objective of the Format Constraint is purely syntactic: to force the model to actively attend to and organize specific visual cues (spatial relationships, key objects, logical structure) to aid its reasoning process.** The model's policy then learns to internalize this method into its core visual-linguistic representations. **The format is a trigger mechanism for acquiring the skill, not the skill itself.** The D2I (Intuitive) mode is successful because **it allows the model to leverage the robust, high-quality skill (i.e., enhanced internal visual attention) acquired during training, while avoiding the generation of the low-quality, failure-prone external output.**
>
> > **W2: If a model could generate the correct image parsing output but still produced the wrong final answer, it would strongly demonstrate the limitations of the D2D paradigm itself, rather than the model's learning ability.**
>
> We agree that analyzing D2D failures is crucial. **We found several cases, particularly in the LOC strategy, where the model correctly identified a reasonable crucial region with the bounding box, but nonetheless proceeded to reach the wrong final answer in its reasoning chain.** We attribute this persistent failure to these factors inherent in the D2D paradigm:
>
> 1. Generating a coordinate (LOC) is an isolated act that directs the model's attention to a region. However, **merely attending to a region does not guarantee that the model will successfully interpret its semantic meaning and apply it correctly within the subsequent reasoning steps.** The ability to truly enhance reasoning relies on the fusion of visual analysis with high-level semantic reasoning—a complex, non-explicit ability. In the D2D paradigm, **the explicit generation of the coordinate often becomes a detour or a symbolic checkpoint that is treated separately from the actual language inference, failing to fully integrate the visual insight into the core reasoning policy.**
>
> 2. In the D2D paradigm, **the model is conditioned on its own generated deliberate steps.** If the model generates a plausible but slightly incorrect parsing or bounding box (which happens often, as these are hard tasks), the subsequent reasoning is hard-constrained to follow this incorrect premise. The model essentially talks itself into a wrong answer.
>
> 3. The D2D paradigm, **by mandating the specific structural output (e.g., <box>...</box> or <parse>...</parse>) at test time, severely constrains the model's capacity for autonomous exploration.** The model is rigidly bound to the self-generated artifact it just produced.  When the model generates a plausible but ultimately suboptimal artifact (like a correct bounding box but the wrong interpretation of that box), the D2D constraint prevents the model from exploring alternative reasoning paths that could ignore the misleading artifact and find the correct solution. **On the contrary, by removing the explicit deliberate tokens at test time, D2I removes this hard conditioning.** The model utilizes the implicit capability (visual understanding) but is free to explore a reasoning path that isn't locked into a specific, potentially erroneous pre-generated structure.
>
> 4. **This is also supported quantitatively by our Pass@k analysis (Figure 4).** D2D models saturate quickly (lower upper bound), indicating that their rigid reasoning paths often lead to dead ends. D2I models show a higher upper bound, confirming they avoid these "self-imposed" traps.
>
> **We have added this part in Appendix M in our paper.**

---

> ### Author Response · Authors · 2025-12-01
> **Response 2**
>
> > **W3: The experiments lack an upper-bound baseline (even a theoretical one) that uses "content rewards".**
>
> We appreciate the reviewer's insistence on establishing a theoretical upper bound using "content rewards" (i.e., Ground Truth supervision). We address this from two perspectives:
>
> 1. **We already included a form of content supervision in our ablation study, as reported in Table 2.** SFT_loc and SFT_par models were trained using on datasets annotated with Ground Truth content (GT coordinates for LOC, GT parsing for PAR). This serves as the direct baseline. Our D2I (RL-Only) method outperforms or matches these SFT baselines. This suggests that **rigidly forcing the model to reproduce human-annotated GT content is suboptimal for complex reasoning. The model benefits more from self-discovery via the Format Reward and GRPO, demonstrating the limitation of the content supervision.**
>
> 2. The alternative interpretation is to use human-labeled content accuracy as the reward signal during the RL training phase. **We did not pursue this for these practical reasons**: (1) The training dataset (GEOQA-8K) currently **lacks the necessary high-quality, granular annotations for content accuracy** (e.g., scoring a bounding box's IoU or a parsing tree's logical correctness) needed for an RL reward function. (2) Generating such high-quality, dense annotations for a large-scale training set is **time-prohibitive and cannot be accomplished within a limited timeline** (e.g., this review cycle). Our D2I method was specifically designed to be annotation-free to overcome this barrier.
>
> In summary, our existing SFT results demonstrate the empirical limits of content supervision. We prioritize a highly efficient, annotation-free D2I approach, which our results show is superior to the costly alternative.

---

### Official Review · Reviewer_kLio · 2025-11-03

**Soundness:** 3
**Presentation:** 4
**Contribution:** 3
**Rating:** 4
**Confidence:** 3

**Summary:**

The paper proposes Deliberate-to-Intuitive (D2I), a training–inference paradigm for multimodal LLMs. During training, the model is guided by deliberate reasoning strategies and rewarded with rule-based format rewards (plus answer correctness), which strengthens modality alignment without extra annotations or complex preference models. At evaluation, the scaffolds are removed and the model responds intuitively, implicitly leveraging what it learned. On visual-math benchmarks (in-domain and out-of-domain), D2I outperforms baselines, suggesting that simple, programmatic format rewards can foster transferable multimodal reasoning and that training-time reasoning depth can be decoupled from test-time response style.

**Strengths:**

1. The core idea directly addresses data and engineering bottlenecks in multimodal reasoning. Because the rewards are deterministic and easy to verify (presence/structure of tags, answer format), the approach is cheap to implement, simple to maintain, and broadly reproducible. This lowers the barrier to deploying reasoning-oriented training at scale and makes the method attractive beyond research prototypes.

2. D2I’s separation between deliberate training and intuitive inference is conceptually tidy and practically useful. The scaffolds force the model to practice grounding and decomposition during training, yet inference discards them, reducing latency, token cost, and brittleness to low-quality intermediate outputs. The result is an appealing operational profile: you pay the complexity during training, then reap simpler, faster, and often more robust inference at deployment.

**Weaknesses:**

1. The format rewards primarily check structure, including shape of <box>, <crucial>, <parse>, rather than semantic fidelity. A box that doesn’t truly cover the evidence, a plausible-sounding justification that is not grounded in the image, or a parse with internally inconsistent relations can still receive a reward. This risks teaching “good-looking” reasoning artifacts that do not guarantee genuine visual grounding.

2. Most compelling results hinge on a single backbone/configuration and visual-math–centric benchmarks. The paper would be stronger with cross-architecture tests (other MLLM families and sizes), variance across random seeds, statistical significance, and detailed compute profiles. Without these, it’s hard to judge how robustly D2I generalizes or what the real cost–benefit curve looks like in production settings.

3. Although test-time scaffolds are removed, prompts are still biased toward stepwise answers rather than truly concise, System-1–style outputs. The work would benefit from a more systematic exploration of response styles and their trade-offs in accuracy, hallucination, and usability, especially for applications that value brevity or strict format constraints at inference.

4. On some general multimodal benchmarks, gains are weaker or mixed, suggesting domain- or scaffold-sensitivity.

**Questions:**

See above weakness

**Details Of Ethics Concerns:**

N.A.

---

> ### Author Response · Authors · 2025-12-01
> **Response 1**
>
> Thank you for reviewing our paper and providing valuable comments. Below are our responses:
>
> > **W1: Teaching good-looking reasoning artifacts don't guarantee visual grounding.**
>
> We thank the reviewer for raising this insightful question. Our core opinion is that the success of D2I relies on its **enhancing the model's visual understanding capacity**, and format reward serves only as a trigger for this process, not the final learning signal.**
>
> The format reward's purpose is to **forcefully compel the model to generate a structured artifact** (e.g., coordinates for LOC or a structural parsing result for PAR). To successfully generate these structures, the model is forced to perform explicit visual-linguistic alignment (e.g., aligning the concept "vertex A" with the correct visual coordinates). **If the model fails to attend to the image, it cannot generate a valid format.**
>
> Also, **whether this generated structure is semantically accurate is determined by the Answer Reward.** If the model generates a syntactically correct but visually ungrounded path (e.g., a coordinate box that misses the evidence), this path will result in a wrong answer and receive a negative reward. The model learns **not just to generate well-formed text, but to generate a well-formed, visually-aligned structure that causally leads to the correct final answer.**
>
> Besides, **if D2I merely taught good-looking artifacts, its performance on OOD tasks would inevitably degrade.** However, **our significant gains on complex OOD benchmarks (MathVista, MathVerse) confirm that D2I successfully internalizes the enhanced visual alignment skill acquired during deliberate training into a robust, generalizable internal representation.** Even when the scaffolding is removed at inference, the model utilizes this enhanced intuitive visual grounding to solve novel, complex problems. In summary, the Format Reward is the means to enforce structure, but the Answer Reward is the end that ensures the learned structure contributes to genuine and generalizable visual understanding.
>
> > **W2: The paper would be stronger with cross-architecture tests.**
>
> We appreciate the reviewer's insistence on generalizability and reproducibility measures. **We have conducted new on Qwen2.5-VL-3B and InternVL2.5-8B, shown in **Appendix K and L in our latest version**. D2I successfully enhances the performance of the smaller 3B model. This confirms that the method is potent enough to boost even models with limited capacity. Applying D2I training to the InternVL2.5-8B architecture results in similar performance uplifts on key  benchmarks. This strongly validates that D2I is a generalizable training paradigm, not a tuning trick specific to the Qwen family. **Meanwhile, the results we report are averaged over five runs, and the improvements from D2I are statistically significant. Our experimental settings have been updated and supplemented in Appendix B.**
>
> | **Method** | **GEOQA-8K** | **MathVerse (mini)** | **MathVista (mini)** | **MATH-Vision (full)** | **MME (sum)** | **MMVet (turbo)** | **MMMU (val)** | **SEED** | **POPE** |
> |-----------|--------------|----------------------|-----------------------|------------------------|---------------|-------------------|----------------|---------|---------|
> | Qwen2.5-VL-3B* | 35.3 | 38.2 | 61.9 | 17.9 | 2102.8 | 60.2 | 52.9 | 62.4 | 71.0 |
> | └─ w/ GRPO | 46.1 | 37.1 | 56.3 | 17.7 | 2154.6 | 57.0 | 53.1 | 60.9 | 69.8 |
> | └─ w/ GRPO† | 46.4 | 37.5 | 57.2 | 16.2 | 2132.5 | 57.2 | 52.4 | 61.2 | 70.3 |
> | └─ w/ D2D_loc (ours) | 48.4 | 42.4 | 61.0 | 19.4 | 2170.2 | **61.7** | 54.0 | 60.8 | 70.0 |
> | └─ w/ D2I_loc (ours) | 50.2 | 43.1 | **64.2** | **20.9** | 2146.3 | 59.9 | 52.6 | 61.9 | 70.4 |
> | └─ w/ D2D_jus (ours) | 40.3 | 40.2 | 60.3 | 18.5 | 2140.8 |  61.3 | 53.8 | 60.2 | 69.5 |
> | └─ w/ D2I_jus (ours) | **52.1** | **43.7** | 63.2 | 17.3 | 2129.0 | 55.7 | 53.1 | 61.0 | 67.6 |
> | └─ w/ D2D_par (ours) | 50.0 | 40.3 | 60.2 | 16.9 | **2231.5** | 60.8 | **56.2** | 64.9 | **73.5** |
> | └─ w/ D2I_par (ours) | **52.1** | 42.3 | 62.7 | 19.1 | 2174.1 | 57.2 | 51.3 | **66.3** | 72.1 |
> | InternVL2.5-8B* | 40.8 | 37.2 | 61.0 | 19.2 | 2298.1 | 61.9 | 54.2 | 64.2 | 84.8 |
> | └─ w/ GRPO | 52.7 | 41.8 | 67.1 | 20.8 | 2319.3 | 64.7 | 59.9 | 67.1 | 88.2 |
> | └─ w/ GRPO† | 57.4 | 45.6 | 67.8 | 19.3 | 2322.8 | 60.4 | 60.1 | 66.4 | 85.9 |
> | └─ w/ D2D_loc (ours) | 54.2 | 44.9 | 69.2 | 21.0 | **2358.2** | 62.8 | 63.7 |69.1 |  89.2 |
> | └─ w/ D2I_loc (ours) | **59.2** | **47.1** | **69.8** | 22.4 | 2273.6 | 63.3 | **64.0** | 68.7 | 88.5 |
> | └─ w/ D2D_jus (ours) | 53.8 | 44.2 | 67.5 | 17.9 | 2301.6 | **65.3** | 62.3 | **69.8** | **89.3** |
> | └─ w/ D2I_jus (ours) | 50.3 | 45.2 | 64.7 | 21.4 | 2326.7 | 61.0 | 61.8 | 66.2 | 82.0 |
> | └─ w/ D2D_par (ours) | 54.0 | 42.9 | 68.1 | 20.4 | 2340.9 | 62.7 | 60.9 | 67.5 | 88.7 |
> | └─ w/ D2I_par (ours) | 57.7 | 45.8 | 69.3 | **23.9** | 2306.3 | 60.4 | 60.4 | 67.9 | 89.0 |

---

> ### Author Response · Authors · 2025-12-01
> **Response 2**
>
> > **W3: The work would benefit from a more systematic exploration of response styles and their trade-offs in accuracy, hallucination, and usability, especially for applications that value brevity or strict format constraints at inference.**
>
> We appreciate the reviewer's focus on the critical balance between accuracy and usability in real-world applications. **However, our design choice to prioritize stepwise answers over truly concise System-1 output is directly motivated by the nature and complexity of the task we are addressing**:
>
> 1. Our work is fundamentally **focused on complex, long-chain reasoning in visual-math domains**. In such complex tasks, Answer-Only (System-1) responses are prone to poor performance because they inherently risk omitting critical intermediate reasoning steps and visual grounding details. Therefore, the core of our research is **finding methods to maximize accuracy through a robust, long reasoning process.**
>
> 2. **The low performance of the Answer-Only style is empirically supported by our Table 1 results: The Qwen2.5-VL-7B (our re-implementation without specific CoT/Deliberate prompting) represents an Answer-Only style baseline.** This baseline consistently yields significantly lower accuracy than D2I and D2D models on both in-domain and out-of-domain benchmarks. This strongly justifies our conclusion that for this class of problems, sacrificing the stepwise output severely compromises the final result.
>
> 3. We agree that the overall utility involves efficiency. **We have controlled the overhead of the stepwise output by limiting the maximum output token length to 1024 tokens.** Empirically, the inference time for D2I (which includes the verbose reasoning path) is, on average, only 1 to 2 seconds slower than the Answer-Only baseline.
>
> Given the substantial performance gain (up to $\sim$14% on complex tasks), the marginal increase in latency (1-2 seconds) is an acceptable trade-off for the complexity of the tasks addressed. Our choice of stepwise output is a necessary and justified design decision to achieve state-of-the-art accuracy in System-2 reasoning, which is the central focus of this paper.
>
> > **W4: On some general multimodal benchmarks, gains are weaker or mixed, suggesting domain- or scaffold-sensitivity.**
>
> We agree that D2I's gains are less pronounced or mixed on general multimodal benchmarks (as observed in our MME and MMMU results). But **this phenomenon is expected and aligns with our core claim, not suggesting domain- or scaffold-sensitivity. We actually addressed this phenomenon in Section 6.1 of our paper, attributing it to the over-divergence of the intuitive reasoning process on general tasks.** As stated in our analysis, For general benchmarks, it may become overly divergent in its reasoning process, thereby missing the correct answers and resulting in a negative impact on overall performance".
>
> **Complex reasoning benchmarks (e.g., MathVista) naturally require multi-step and abstract reasoning**, symbolic manipulation, and long-chain deduction. These tasks benefit from a broad reasoning search space to explore diverse solution paths, which aligns well with D2I’s flexible reasoning style. In contrast, **general multimodal benchmarks (e.g., MME) are much simpler**—they mainly depend on precise visual understanding and direct question answering, and thus do not require extensive deliberation.
>
> **As discussed in Section 6.4, removing format constraints in D2I increases entropy and encourages exploration. While useful for hard reasoning tasks, this becomes counterproductive on simpler benchmarks**: the model’s reasoning drifts, over-expanding what should be a straightforward recognition task and reducing accuracy. **Conversely, D2D’s enforced structure (e.g., producing a fixed parsing format) serves as a strong regularizer, keeping outputs focused.** This makes D2D more suitable for tasks dominated by visual perception rather than composite reasoning.

---

### Official Review · Reviewer_a5bU · 2025-11-03

**Soundness:** 3
**Presentation:** 2
**Contribution:** 3
**Rating:** 4
**Confidence:** 3

**Summary:**

The paper introduces D2I framework to enhance the reasoning capabilities of MLLMs. The main idea is to train models to engage in deliberate, structured reasoning during training and switch to more flexible, intuitive reasoning during inference. This paper proposes three different strategies and only the reponse format is supervised in the training stage. Experiments results on on multiple benchmarks such as MathVerse and MathVista show that D2I framework can surpass D2D in most cases.

**Strengths:**

1. This paper proposes three different reasoning strategies, which helps model to learn the alignment across image and text modals. Also the training stage focuses on format reward, which means additional human annotations or expensive data generation are not required, making the method cost-effective and scalable.
2. Allowing flexible generation in inference stage is innovate, and the experiments results on in-domain dataset further support the effectiveness of this D2I framework.

**Weaknesses:**

1. It looks like the citation format is somehow wrong. There are no brackets warping the citations, makes them mixed with other texts. Also, the figures are not well organized, for instance Fig 6 shows before Fig 5.
2. The results on out-of-domain dataset shows that D2D achieve better performance than D2I models. However authors lack detailed description of this phenomenon.

**Questions:**

1. The results on general OOD dataset like MME shows that D2D can achieve better results than all of the three D2I models. What may be the reasons?
2. I'd like to know more about the the PAR strategy. For geometry tasks the textual explanation (JUS) and structrual parsing (PAR) may show great difference. However on other general VQA tasks how does PAR works? Especally there are no additional annotations to help the model learn GT structure language.

---

> ### Author Response · Authors · 2025-12-01
> **Response 1**
>
> Thank you for reviewing our paper and providing valuable comments. Below are our responses:
>
> > **W1: It looks like the citation format is somehow wrong.**
>
> This has been rectified.
>
> > **W2 & Q1: Out-of-domain dataset shows that D2D achieve better performance than D2I. Lack detailed description.**
>
> We appreciate the reviewer pointing this out. **We actually addressed this phenomenon in Section 6.1 of our paper**, attributing it to the over-divergence of the intuitive reasoning process on general tasks. As stated in our analysis, For general benchmarks, it may become overly divergent in its reasoning process, thereby missing the correct answers and resulting in a negative impact on overall performance".
>
> To further analysis, the D2I framework removes the rigid format constraints during inference to encourage exploration (higher entropy). **While this exploration is crucial for solving complex, multi-step mathematical problems by helping the model escape local optima, it can be detrimental for general visual tasks like MME or POPE.** For complex reasoning Tasks (e.g., MathVista), they are inherently difficult, requiring multi-step, long-chain, composite reasoning, abstract symbol manipulation and so on. These tasks demand a wide reasoning search space to explore potential solutions and avoid local optima. D2I's high-entropy, flexible reasoning is optimized for this domain. **However, for general multimodal tasks (e.g., MME), they are generally simpler, often relying on accurate visual recognition and direct QA. They do not necessitate extensive deliberation and benefit more from focused, concise outputs.**
>
> As we analyzed in Section 6.4, removing format constraints in D2I is designed to increase entropy, which aids exploration. However, this exploration becomes a disadvantage on simpler general tasks. **D2I's flexibility causes its reasoning process to become overly divergent when faced with simple recognition problems. The model unnecessarily complicates the task, causing the reasoning path to drift away from the straightforward, correct answer.** In other words, D2I's enhanced capacity for composite reasoning can sometimes negatively impact its performance on tasks that only require direct visual perception. In contrast, D2D's compulsory structured output (e.g., generating a parsing structure) acts as a powerful regularizer during inference. This format constraint keeps the model's output focused and concise, making it better suited for tasks that rely primarily on accurate visual recognition.

---

> ### Author Response · Authors · 2025-12-01
> **Response 2**
>
> > **Q2: I'd like to know more about the the PAR strategy. For geometry tasks the textual explanation (JUS) and structrual parsing (PAR) may show great difference. However on other general VQA tasks how does PAR works? Especally there are no additional annotations to help the model learn GT structure language.**
>
> Thank you for this highly insightful comment. **The formal, structural parsing required for geometry tasks (e.g., points, lines, angles) is not directly applicable to general VQA and scene understanding tasks.** For general VQA tasks, we adopted a mechanism that **serves the same functional purpose as geometric PAR, which is to compel the model to perform structured visual-to-linguistic alignment but adapted for general scenes.** The goal of PAR in geometry is to ensure the model thoroughly understands the diagram's structure. For non-geometric images, we force the model to articulate the **key objects and their spatial** and semantic relationships using a structured language. The prompt is:
> ```
> You FIRST parse the image with structure language, and then think about the reasoning process as an internal monologue and finally provide the final answer. The parsing result MUST BE enclosed within <parse> </parse> tags. The reasoning process MUST BE enclosed within <think> </think> tags. The final answer MUST BE enclosed within <answer> </answer> tags. Note the parsing result is parsing the image with listing all the objects and their relationships using predicate format. The overall response format is <parse> parsing result here </parse> <think> reasoning process here </think> <answer> final answer here </answer>.
> ```
>
> For the GT in PAR setting, we **do not require additional annotations because we leverage the pre-existing zero-shot capabilities of the base Multimodal LLM**. Our preliminary tests confirmed that Qwen2.5-VL possesses an inherent ability to generate descriptive, structured parsing results for arbitrary images when prompted correctly. The Format Reward simply reinforces the structural syntax, while the GRPO Final Answer Reward compels the model to generate semantically meaningful parsing that leads to the correct answer. The model learns to discover its own useful structure rather than mimic an external GT. **Furthermore, the results in Table 2 validate the flexibility and superiority of our designed approach without specific GT.** We conducted an experiment (SFT_par followed by SFT-RL_par) where we first fine-tuned the model on ground-truth parsing datasets. The results were not ideal and in many cases, performance was lower than our purely self-discovered RL-Only method. This crucial ablation demonstrates that the base model already possesses the necessary latent parsing capability, and our D2I approach is more effective at activating and refining this self-discovered capability than forcing it to conform to external supervision.

---

### Official Review · Reviewer_TyaT · 2025-11-05

**Soundness:** 2
**Presentation:** 2
**Contribution:** 2
**Rating:** 4
**Confidence:** 4

**Summary:**

This paper proposes a Deliberate-to-Intuitive (D2I) reasoning framework designed to enhance the understanding and reasoning capabilities of Multimodal Large Language Models (MLLMs). Specifically, the authors introduce three training strategies—Region Localization (LOC), Region Justification (JUS), and Parsing Consistency (PAR)—which are applied during training but removed during inference to strengthen the model’s reasoning ability. Experimental results demonstrate that D2I achieves improvements on both in-domain and out-of-domain benchmarks.

**Strengths:**

1. This paper is straightforward and easy to understand.
2. The three methods sound reasonable for the training stage.
3. The results achieve improvements on both in-domain and out-of-domain benchmarks.

**Weaknesses:**

1. Adding citations in Line 094-096 about how deliberate reasoning behaviors are commonly adopted can help readers gain a better understanding of this field.
2. The citation at Line 499 is incomplete.
3. Line 394: MMe -> MME
4. Lacks citations and discussion of previous reasoning MLLMs and related works, e.g., [1-5]

[1] Vision-r1: Incentivizing reasoning capability in multimodal large language models

[2] R1-vl: Learning to reason with multimodal large language models via step-wise group relative policy optimization

[3] Video-R1: Reinforcing Video Reasoning in MLLMs

[4] Seg-zero: Reasoning-chain guided segmentation via cognitive reinforcement

[5] Vlm-r1: A stable and generalizable r1-style large vision-language model

**Questions:**

1. What about the results of I2I learning? Is it equivalent to common GRPO?
2. What is the value of k in Figure 4?
3. Entropy increases means randomness also increases, so how is reproducibility ensured? What is the fluctuation range of the results?
4. Providing inference-time hyperparameters (e.g., temperature, top-p/top-k, and other relevant settings) is important for replication.
5. The authors state that D2I leads to a higher entropy distribution, implying greater randomness in the model’s outputs. Could this be due to the difference between training prompts and inference prompts? Additionally, how do the authors interpret this increase in entropy — is higher entropy during inference considered beneficial or potentially harmful?

---

> ### Author Response · Authors · 2025-12-01
> **Response 1**
>
> Thank you for reviewing our paper and providing valuable comments. Below are our responses:
>
> > **W1: Adding citations in Line 094-096 about how deliberate reasoning behaviors are commonly adopted can help readers gain a better understanding of this field.**
>
> While the current literature may not feature explicit frameworks that perfectly align with our novel Deliberate-to-Intuitive (D2I) distinction or its dual reasoning concepts, we have intentionally selected analogous and highly related research work to firmly ground our methodology within the recognized landscape of advanced reasoning behaviors. We chose to specifically highlight [1] and [2] to effectively contextualize our Deliberate Reasoning strategies (LOC, JUS, PAR) within the broader MLLM literature:
>
> 1. [1] is a seminal example of integrating multimodal information directly into the explicit reasoning chain. It supports the foundational concept that for complex multimodal tasks, deliberate behavior must involve the cross-modal alignment of visual cues and language. Our LOC, PAR, and JUS strategies are tailored manifestations of this principle, forcing specific, structured alignments during training. **Citing this paper clearly positions our work as an advancement built upon the established Multimodal CoT framework.**
>
> 2. [2] represents an advanced form of deliberate reasoning that involves meta-cognition and refinement. By citing a method focused on enhancing the model's intrinsic ability to verify and correct its own thought process, we demonstrate that "Deliberate Reasoning" encompasses not only step-by-step generation but also high-level, internal reflection aimed at unlocking deeper reasoning skills. **This broadens the scope of our "Deliberate" training phase.**
>
> [1] Multimodal Chain-of-Thought Reasoning in Language Models.
>
> [2] Boosting LLM Reasoning via Spontaneous Self-Correction
>
> > **W2 & W3: The citation at Line 499 is incomplete. Line 394: MMe -> MME**
>
> These have been rectified.

---

> ### Author Response · Authors · 2025-12-01
> **Response 2**
>
> > **W4: Lacks citations and discussion of previous reasoning MLLMs and related works**
>
> We sincerely thank the reviewer for pointing out this critical omission regarding the established literature on Rule-based Reinforcement Learning. The cited works are highly relevant as they pioneered and demonstrated the utility of the R1-style learning paradigm using rule-based rewards to enforce structured behavior. **Our discussion with these works and our framework is as below:**
>
> **Vision-R1** [3] introduced the core R1 paradigm that uses rule-based rewards to incentivize MLLMs to generate structured reasoning paths. It proved that format constraints can effectively enforce deliberate behavior and teach new skills without human content supervision. But the rigid requirement for structured output at inference time severely suppresses exploration and introduces brittleness, leading to suboptimal performance on OOD tasks compared to D2I.
>
> **R1-VL** [4] refined the R1 framework by introducing GRPO, focusing on step-wise relative quality feedback to optimize the policy. It provided the highly efficient and stable GRPO that D2I's deliberate training phase directly utilizes, significantly accelerating skill acquisition. But it is a coupled D2D model. The enhanced reasoning skill learned efficiently through GRPO is still constrained by the format requirement at inference. D2I unlocks this skill by removing the constraint.
>
> **Video-R1** [5] applied the R1 paradigm to video reasoning tasks, designing temporal-specific rule rewards to enhance multi-frame, deliberate processing in MLLMs. It validated the cross-modal generalizability of the R1 concept to complex sequence modalities like video. However, video reasoning is resource-intensive. Mandating structured D2D output exacerbates latency and inflexibility. D2I's principle of intuitive (concise) inference is even more crucial here to maintain efficiency and generalizability.
>
> **Seg-Zero** [6] utilized R1-style reinforcement to guide and improve non-linguistic, structured visual output via a reasoning chain. It demonstrated that format reinforcement can drive the generation of structured outputs that are not pure text, linking thought chains to low-level visual perception tasks. But its D2D nature is necessary for the task. D2I's advantage is specific to reasoning tasks where the structure is an intermediate scaffold. D2I’s success lies in strategically discarding the scaffold to maximize reasoning flexibility, a strategy Seg-Zero cannot adopt.
>
> **VLM-R1** [7] focused on creating a stable and generalizable R1-style VLM by addressing common instabilities in RL training, greatly improved the training stability and scalability of the R1 framework, providing a more robust foundation for any subsequent deliberate training. Despite improving generalization in training, it remains a coupled D2D model. D2I proves that true OOD robustness is achieved not merely by stabilizing the training, but by strategically decoupling the acquired skill from the constraining structure at inference time.
>
> **We have added this part in the Appendix J in our paper.**
>
> [3] Vision-r1: Incentivizing reasoning capability in multimodal large language models
>
> [4] R1-vl: Learning to reason with multimodal large language models via step-wise group relative policy optimization
>
> [5] Video-R1: Reinforcing Video Reasoning in MLLMs
>
> [6] Seg-zero: Reasoning-chain guided segmentation via cognitive reinforcement
>
> [7] Vlm-r1: A stable and generalizable r1-style large vision-language model
>
> > **Q1: What about the results of I2I learning? Is it equivalent to common GRPO?**
>
> We appreciate the opportunity to clarify these definitions. **No, the Common GRPO baseline is not equivalent to I2I; rather, it represents a standard form of D2D.** In our experiments, the **w/ GRPO baseline is trained with a specific format reward that enforces a structured Chain-of-Thought (enclosed within <think> and <answer> tags).** During evaluation, it is also prompted to generate this specific format. Therefore, **Common GRPO falls under the D2D paradigm**: it is trained deliberately and acts deliberately. In Table 1, we compare our proposed D2I against the GRPO (D2D) and GRPO \dagger (D2I, because we remove the format requirement in the output) baselines. The results show that D2I outperforms Common GRPO, indicating that decoupling the rigorous training format from inference (D2I) is superior to the standard coupled approach.
>
> > **Q2: What is the value of k in Figure 4?**
>
> In Figure 4, **the value of $k$ (representing the number of samples for Pass@k evaluation) ranges from 1 to 64**. The x-axis represents the number of samples $k$, specifically plotting discrete values at intervals (e.g., $k=1, 4, 8, \dots, 64$). The curves consistently show that D2I tends to maintain a higher upper bound (Pass@k performance) compared to D2D as $k$ increases, indicating that D2I's broader search space effectively captures the correct solution more often.

---

> ### Author Response · Authors · 2025-12-01
> **Response 3**
>
> > **Q3: How is reproducibility ensured? What is the fluctuation range of the results?**
>
> This is an insightful question. **The increase in entropy is indeed a designed feature of D2I to encourage exploration, but reproducibility is strictly controlled via experimental protocols.**
>
> To ensure reproducibility, we **fixed the random seed** during both training and inference. We also used a **fixed sampling temperature (Temperature = 1) for all comparative experiments to ensure fair comparisons.**
>
> For the fluctuation range of the results, we clarify that **the results reported in Table 1 are the average of 5 independent runs**. In our experiments, we observed that the **variance across these runs is approximately 0.2% to 2%**. However, **due to the readability of the table in the paper, we choose not to display the range of fluctuation. **
>
> Despite the increased entropy, the performance improvement of D2I over the baselines remains statistically significant. **The higher entropy in D2I indicates a broader exploration of the reasoning space rather than randomness in the final answer.** This "exploration" is precisely what allows D2I to avoid the local optima that trap the more deterministic D2D models.
>
> > **Q4: Providing inference-time hyperparameters.**
>
> The inference-time hyperparameters: Sampling Temperature: 1. Max Response Length: 1024 tokens. Prompt: We utilize the specific inference prompts detailed in Appendix C (Inference Stage). Hardware: All experiments are conducted on NVIDIA A100 GPUs. **We have updated these settings in Appendix B of our paper.**
>
> > **Q5: Could this be due to the difference between training prompts and inference prompts? How do the authors interpret this increase in entropy — is higher entropy during inference considered beneficial or potentially harmful?**
>
> About Cause of Entropy Increase: the increase in entropy is largely driven by the removal of the formatting constraints (the Deliberate prompts) during inference. **As we hypothesize in the paper, training constrains the response search space to force skill acquisition, while inference removes these constraints. The Token Distribution Shift analysis (Figure 5) confirms that D2I actively restructures the output space rather than just being a passive artifact.**
>
> We interpret this high entropy as beneficial and critical for unlocking reasoning potential:
>
> 1. **The lower entropy in D2D models suggests they are often over-fitted to the rigid training format.** D2I's higher entropy indicates a more exploratory policy that can bypass these rigid paths to find correct answers.
>
> 2. This is supported by our Pass@k results (Figure 4). If the higher entropy were harmful (i.e., pure noise), the Pass@k curves would not show much improvement. Instead, **D2I shows a higher upper bound than D2D in the most situations, proving that the increased diversity is effectively covering the correct reasoning paths.**
>
> 3. As seen in Figure 3, models with rigid formats (GRPO \dagger) often produce similar but incorrect reasoning traces. **The Intuitive mode (e.g., $D2I_{jus}$) allowed the model to diverge and identify the correct concept, leading to the right answer.**

---

### Meta-Review · Area_Chair_VuKQ · 2026-01-02

**Summary:**

The submission receives initial scores of 4, 4, 4, 6, and 6. The AC has carefully read the paper, along with all reviews and rebuttals. The AC finds that the D2I idea is interesting and has potential, possibly serving as a means for scalable training. However, the main concerns remain that the true reasons why D2I works are unclear and that a complete, systematic evaluation is lacking. Based on the current version, the AC recommends rejection. Nevertheless, the AC recognizes the potential of the work and encourages the authors to revise the paper based on the reviewers’ comments.

More specifically,

1. The authors provide many explanations, but they often appear subjective or inconsistent. While several figures are presented, they offer indirect evidence and speculation rather than fully supporting the conclusions. The AC believes these hypotheses may be valid but require rigorous experimental validation. For example,
    1. Lower entropy does not directly imply overfitting to a rigid format, and similarly, higher entropy does not necessarily indicate improved inference performance. While it may reflect greater exploration, it could also increase confusion due to lower confidence or other factors.
    2. The rebuttal claims that the success of D2I relies on enhancing the model's visual understanding capacity. However, if this were the main factor, one would expect similar improvements on general benchmarks. The rebuttal's explanation for the limited gains on general benchmarks, that this exploration is crucial for solving complex, multi-step mathematical problems, appears somewhat inconsistent with the claim about visual understanding and suggests a tension in the argument.
    3. The discussion about format rewards versus content rewards is important. The AC believes that the failures of D2D highlight potential issues with format rewards, which deserve detailed analysis. Furthermore, showing how D2I partially overcomes these issues, with sufficient supporting evidence, could provide a key explanation for why D2I works.
2. The rebuttal adds experiments on Qwen-2.5-VL-3B and InternVL-2.5-8B, but a more systematic evaluation as suggested by the reviewers, particularly regarding the generalization of D2I, would strengthen its contributions and significance.
3. The performance improvements are mixed across different reward formats and less effective on general benchmarks. These issues require better explanation, or alternatively, an analysis of whether they can collectively lead to consistent gains.

**Reviewer Concerns:**

The reviewers raised several concerns, with the key and common issues summarized below.

1. Reviewer TyaT, a5bU, kLio, NXGx, and Dp4P: raise questions about the explanation and claims regarding why D2I works.
    1. kLio: notes that the format reward does not guarantee correctness of content such as grounding. The rebuttal explains that the success of D2I relies on enhancing the model's visual understanding capacity, and that the format reward serves only as a trigger, not as the final learning signal.
    2. TyaT: questions the explanation and rationale for the higher entropy distribution of D2I. The rebuttal clarifies that the increase in entropy is largely due to the removal of formatting constraints, and that the resulting higher diversity effectively covers the correct reasoning paths.
    3. a5bU: observes that D2D achieves better performance than D2I on out-of-domain datasets. The rebuttal attributes this to over-divergence of the intuitive reasoning process on general tasks.
    4. NXGx: asks why forcing the model to output coordinates (LOC) during training improves performance on intuitive reasoning tasks that do not require coordinates. The rebuttal states that this improvement comes from the internalization of visual-semantic alignment during training.
    5. NXGx: notes the limitations of the D2D paradigm itself. The rebuttal discusses several possible explanations.
    6. Dp4P: questions whether the format rewards are too domain-specific. The rebuttal argues that D2I is not limited to structured tasks and provides some supporting results.
2. Reviewer kLio and a5bU: note that the improvements of D2I on general benchmarks are weaker or mixed.
3. Reviewer kLio, Dp4P: suggest more systematic evaluation to support the conclusions, including cross-architecture tests, random seeds, trade-offs in accuracy, hallucination, and usability, as well as variations in model sizes and training datasets. The rebuttal adds experiments on Qwen-2.5-VL-3B and InternVL-2.5-8B.
4. Reviewer TyaT and a5bU: note issues with missing citations or formatting. The rebuttal addresses these concerns by correcting the citations and adding discussion of related work.
5. Reviewer TyaT: requests clarification and implementation details, such as the choice of k in Figure 4, reproducibility, and inference-time hyperparameters. The rebuttal provides these details and explanations.

The AC considers that the most critical concerns, namely the true reasons why D2I works and the need for more systematic evaluation, have not been fully addressed (see Summary).

**Reviewer Scores:**

Reviewers TyaT and a5bU have their citation concerns addressed, but the explanation for why D2I truly works has not been fully resolved. The AC expects that they will likely maintain their scores, although Reviewer TyaT may increase their score to 6.

Reviewer kLio’s concerns regarding format rewards, more systematic evaluation, and mixed performance have not been fully addressed, and the AC expects the score to remain at 4.

Reviewer NXGx may maintain or lower their score to 4, given the lack of empirical evidence for why intuitive reasoning does not require coordinates and the unresolved questions about format rewards versus content rewards.

Reviewer Dp4P appears less confident, and the AC expects the score to likely remain at 6.

---

### Decision · Program_Chairs · 2026-01-26

Reject